# SuperMIL: Supervised Multiple Instance Learning for Time Series Classification

## Abstract

Decision-making requirements in fields like industrial monitoring and healthcare are equally critical for ensuring the accuracy and interpretability of time series classification (TSC) methods. Multiple instance learning (MIL) with interpretability is a promising framework that decomposes the sequences (bag) into instance-level segments and evaluates their contributions via attention mechanisms. However, the existing MIL methods introduced in the TSC field have two major limitations: 1) degraded instance-level feature learning due to optimizing for bag-level predictions and 2) insufficient modeling of causal dependencies among temporally ordered instances. This paper proposes supervised multi-instance learning (SuperMIL), which iteratively optimizes instance-level pseudo-supervision (from bag-inherited labels) and bag-level weak supervision, yielding discriminative instance features and robust bag-level predictions. Moreover, SuperMIL integrates a Hawkes pooling module and a coupled multi-instance loss. The former captures local inter-instance causality by decomposing excitations into directional similarity and instance differences, and the latter models inter-instance as well as collective loss-global attention interactions to align instance-level and bag-level objectives, both of them synergizing capture local causal causality and global instance semantics. The SuperMIL framework enhances performance in representative TSC models, outperforming traditional MIL methods, as validated through experiments on the UCR and UEA datasets. Code is available at this repository: https://anonymous.4open.science/r/SuperMIL.

## 1 Introduction

Time series (TS) is a sequence of data points ordered chronologically and is commonly observed in fields such as finance, healthcare, and industrial monitoring Rezvani et al. (2019); Wang et al. (2022); Zhang et al. (2023); He et al. (2025); Li et al. (2023); Xiao et al. (2024). Time series classification (TSC) aims to assign category labels to TS data and has become a highly-regarded TS mining task due to its diverse application scenarios Middlehurst et al. (2024); Foumani et al. (2024); Fawaz et al. (2019); Zhang et al. (2024); Wang et al. (2024). As tasks become increasingly complex, the demand for explaining the decision-making processes of classification models has grown significantly Fisch et al. (2011); Theissler et al. (2022). According to the granularity of explanation, existing TSC algorithms are categorized into whole series-based, subsequence-based, and point-based approaches Ismail-Fawaz et al. (2023); Zerveas et al. (2021); Fawaz & et al. (2020); Wang et al. (2017); He et al. (2022); Li et al. (2022); Guillaume et al. (2022); Tan et al. (2022); Schäfer & Leser (2023); Le et al. (2024); Ismail-Fawaz et al. (2022). Representatively, multiple instance learning (MIL) is a typical point-based method that obtains point interpretability by emphasizing the scarcity of discriminative features.

MIL extracts per-time-point features (MIL instances) and treats all TS instances as a 'bag' Early et al. (2023); Chen & et al. (2024). The inherent difficulty in acquiring instance labels Wang et al. (2022); Shao et al. (2023) directs most existing MIL research towards constructing more robust pooling layers that aggregate unsupervised instance features for bag-level prediction Carbonneau et al. (2018); Xiong et al. (2021). The definition of pooling operations and loss functions at the bag level alone results in each instance being only weakly supervised Ouyang et al. (2024); Ma et al. (2024); Qu et al. (2024); Ilse et al. (2018). Consequently, the suboptimal learning of instance-level features Zhang et al. (2024); Liu et al. (2024); Cha et al. (2021); Wang et al. (2023) results in

their poor separability, as illustrated in Fig. 1A. Some studies generate pseudo-labels through label inheritance and propose iterative pseudo-label optimization to enhance instance feature learning Qu et al. (2024). However, these methods typically suffer from high computational complexity and ineffectively leverage bag-level information for guiding instance learning. Furthermore, a pivotal yet frequently overlooked MIL consideration is that pooling operation order dictates the efficacy of parameter optimization for feature extractors and classifiers, where early pooling leads to inferior model optimization and substantially degrades instance learning.

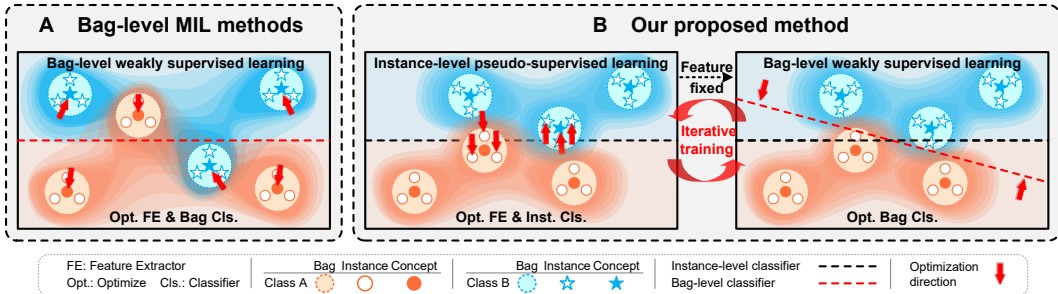

Figure 1: Our iterative co-optimization tackles instance ambiguity and underutilized information in weakly supervised learning, enhancing instance discriminability via instance-level supervision and boosting bag-level classification by aggregating predicted probabilities to minimize information loss.

TS data have an inherent sequential nature, and many time-series problems depend critically on instance ordering (for example, an early anomaly may trigger a later failure) Ruiz et al. (2021); Middlehurst et al. (2024); Liu et al. (2024); Foumani et al. (2024). The standard MIL pooling method has difficulty capturing this structured temporal information. Although TimeMIL has introduced a time-aware pooling mechanism, it primarily focuses on capturing correlations between instances rather than modeling causal relationships Chen & et al. (2024). In contrast, Hawkes processes provide a principled way to model the self-exciting effect of event sequences Zhang et al. (2024); Sun et al. (2022); Yu et al. (2022); Li & Sun (2023); Zhou et al. (2023); Yoon et al. (2023); Shi & Cartlidge (2022); Gao et al. (2024). However, since existing TSC methods lack appropriate instance-level supervision, Hawkes-style excitation modeling has not been integrated into the MIL-based TSC framework.

In summary, existing MIL-based TSC methods have two major limitations: 1) degraded instance-level feature learning and 2) insufficient modeling of causal dependencies among temporally ordered instances. Motivated by these observations, a novel MIL-based TSC framework, supervised multi-instance learning (SuperMIL), is proposed. SuperMIL iteratively optimizes instance-level pseudo-supervision (from bag-inherited labels) and bag-level weak supervision learning, yielding discriminative instance features and robust bag-level predictions, as shown in Fig. 1B. Moreover, SuperMIL integrates a Hawkes pooling module and a coupled multi-instance loss, synergizing capture local causal causality and global instance semantics. The former captures local causality by decomposing inter-instance excitations into causal validity, characterized by directional similarity, and causal strength, characterized by instance differences. The latter models inter-instance as well as collective loss-global attention interactions to align instance-level and bag-level objectives.

The contributions of this work are summarized as follows:

- A novel MIL framework, SuperMIL, is proposed. It mitigates instance-level learning degradation by iteratively optimizing both instance-level pseudo-supervision and bag-level weak supervision learning. Simultaneously, SuperMIL aggregates predictions in the probability space, reducing information loss and achieving robust bag-level predictions.

- A Hawkes pooling module is designed to capture local causality by decomposing inter-instance excitations into causal validity, characterized by directional similarity, and causal strength, characterized by instance differences.

- A coupled multi-instance loss is proposed to align instance-level and bag-level objectives by models inter-instance as well as collective loss-global attention interactions.

The remainder of this paper is organized as follows: Section 2 reviews relevant concepts in TSC and analyzes the theoretical advantages of SuperMIL over GAP and MIL. Section 3 presents an overview of the proposed SuperMIL. Section 4 presents experimental results and concludes the paper in Section 5.

## 2 THEORETICAL ANALYSIS

This section presents the formal definition of TSC and provides a theoretical comparison of SuperMIL with MIL and GAP methods.

### 2.1 PROBLEM FORMULATION

TSC refers to assigning labels to TS data, which can be divided into univariate and multivariate cases. In the univariate task, each time point corresponds to a single value; in contrast, each time point in the multivariate case is represented by a $W$-dimensional vector. Specifically, let $\{X_1, \ldots, X_n\}$ denote a dataset of $n$ multivariate time series, where each $X_i = \{x_i^1, \ldots, x_i^T\}$ consists of $T$ time points, and each $x_i^t \in \mathbb{R}^W$. Given a training set $\{(X_1, y_1), \ldots, (X_n, y_n)\}$, where $y_i$ is the label for $X_i$, the goal is to learn a function $f : X \to Y$ that maps time series data to the label space.

### 2.2 SUPERMIL COMPARE WITH MIL AND GAP

MIL for TSC faces a fundamental trade-off: optimizing solely for bag-level prediction degenerates instance-level representations, while focusing only on instance-level supervision cannot ensure effective bag-level classification. Although recent methods attempt to balance these objectives, most neglect the fact that the primary information loss stems from the timing of the pooling operation. To address this, the SuperMIL framework is proposed, and its theoretical advantages over MIL and GAP are analyzed in terms of information retention with respect to pooling timing. For clarity, attention-based pooling is excluded, and the discussion is limited to the average pooling of the above three paradigms.

SuperMIL employs a dual-path iterative optimization framework that decouples instance-level and bag-level training, as illustrated in Fig. 2. The instance-level branch (left) updates the feature extractor and instance classifier using instance-level loss functions from bag labels as pseudo-supervision (in the absence of instance-level ground truth), enhancing feature discriminability. With parameters frozen, the feature extractor outputs features to the bag-level classifier (right), where instance prediction probabilities are computed and then aggregated in the probability space to produce the bag-level prediction. By postponing aggregation until the final stage, critical information is better preserved throughout both training paths.

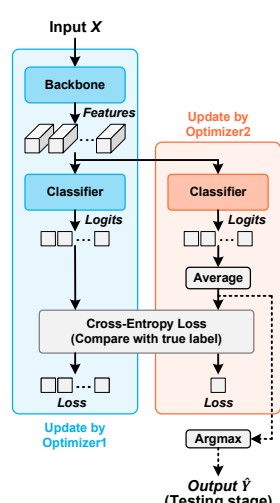

Figure 2: The SuperMIL framework.

For a fair comparison, all paradigms are constructed upon a common backbone comprising a feature extractor $\psi_E$ and a classification head $\psi_C$. The key distinction among GAP, MIL, and SuperMIL methods lies in the step at which average pooling is applied, as illustrated in Fig. 3. To theoretically underpin the intuition about the negative impact of early pooling, a gradient analysis is conducted to reveal key differences in the granularity of parameter updates among GAP, MIL, and SuperMIL. The gradient update mechanisms for each framework are summarized as follows:

- **GAP:** The classification head is updated based on gradients with respect to the pooled global feature vector, i.e., $\frac{\partial \psi_C(\theta_C, \bar{e})}{\partial \theta_C}$.

- **MIL:** Gradients are computed for each instance feature, i.e., $\frac{\partial \psi_C(\theta_C, e_i)}{\partial \theta_C}$, but a uniform weighting coefficient derived from the overall bag prediction error, $p - \mathbf{1}_y$ is applied.

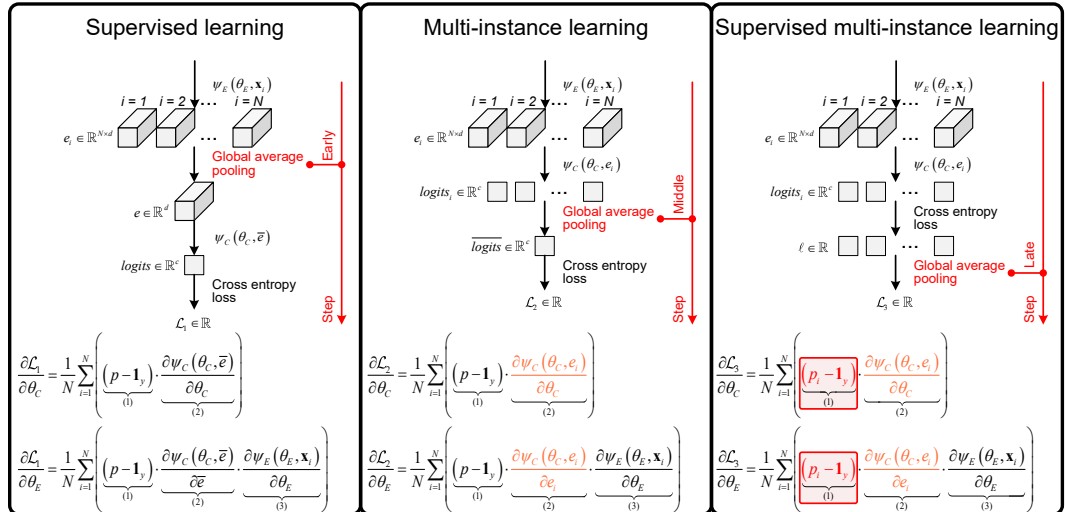

Figure 3: A comparative analysis of GAP-based, MIL-based, and SuperMIL methods, examining the impact of the average pooling step on feature extractor and classifier optimization.

- **SuperMIL:** Instance-specific weighting coefficients, $p_i - \mathbf{1}_y$, are assigned based on each instance's individual prediction error. This approach enables greater emphasis on informative or misclassified instances, thereby enhancing instance-level discrimination.

In this context, $p = \text{softmax}\left(logits\right)$ represents the predicted probability at the bag level, while $p_i = \text{softmax}\left(logits_i\right)$ represents the predicted probability at the instance level.

## 3 METHODOLOGY

Informative instance representations are learned via iterative optimization in the SuperMIL framework, and instance logits from a bag-level classifier are used for final prediction. To address label noise and improve stability, the SuperMIL framework integrates a coupled multi-instance loss for local-global objective alignment and a Hawkes pooling module for modeling temporal causality, as illustrated in Fig. 4 and Algorithm 1.

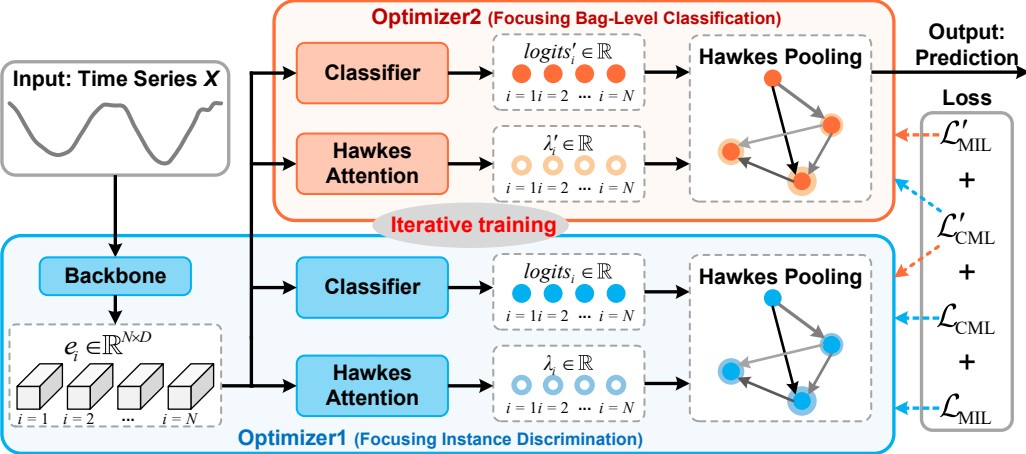

Figure 4: SuperMIL incorporates a Hawkes pooling module to model temporal causality and a coupled multi-instance loss to align bag-instance objectives.

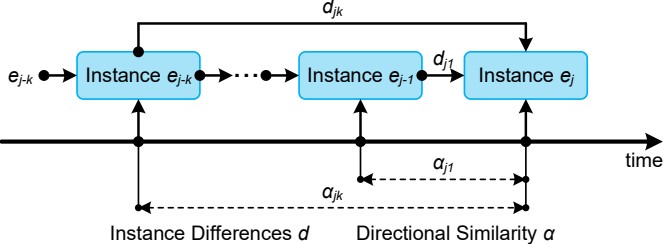

Figure 5: Hawkes attention employs dynamic gating, integrating directional similarity and instance differences, to capture local temporal causality collaboratively.

## 3.1 HAWKES POOLING

Structured aggregation is crucial for preventing information loss and capturing causal dynamics in the MIL-based TSC algorithm. To address this challenge, the Hawkes pooling module, inspired by the Hawkes process Zuo et al. (2020); Zhang et al. (2020), models causal effectiveness through directional similarity and quantifies causal strength using instance differences, as illustrated in Fig. 5.

The causal strength between instances is defined as

$$d_{ji} = \psi_{Di}(e_j - e_{j-i}), \ i \in \{1, 2, \ldots, k\} \tag{1}$$

where $e_i \in \mathbb{R}^D$, with $D$ representing the feature dimension size, denotes the feature representation of the $i$-th instance, $d_{ji} \in \mathbb{R}$ measures the excitation intensity from the $k$-th preceding instance.

The directional similarity is computed as

$$z_{ji} = \frac{\psi_A(e_j) \cdot \psi_A(e_{j-i})}{\sqrt{D}}, \ i \in \{1, 2, \ldots, k\} \tag{2}$$

where $z_{ji} \in \mathbb{R}$ measures the similarity between instances, and $\alpha_{jk} = \frac{z_{jk}}{\sum_{i=1}^{k} z_{ji}}$ is the normalized similarity to the $k$-th previous instance within a window of size $k$.

Dynamic gating within local windows integrates similarity and causal strength to generate incentive weights. This approach prioritizes strong local causal links and suppresses long-range interference.

$$\lambda_j = \sum_{i=1}^{k} \alpha_{ji} \cdot d_{ji} + s_j \tag{3}$$

where $s_i = \psi_S(e_i)$ denotes the self-importance term. For classification adjustment, the resulting causal effect $\lambda_j$ is used to weight the logits ($logits_i \in \mathbb{R}^C$, where $C$ is the number of classes) of each instance. The final bag prediction $p \in \mathbb{R}^C$, is then computed as $p = \text{softmax}(\sum_{i=1}^{N} \lambda_i \cdot logits_i)$.

In summary, the Hawkes attention mechanism can be formulated as $\psi_H = F\{\psi_S, \{\psi_{Di}\}_{i=0}^{k}\}, \psi_A\}$, where $F$ represents a composite mechanism. This strategy captures both correlation and evolutionary dynamics, enabling effective temporal aggregation and enhancing robustness for complex TSC tasks. The specific architectures of all neural modules $(\psi_S, \{\psi_{Di}\}_{i=0}^{k}, \psi_A)$ are detailed in Appendix C.

## 3.2 COUPLED MULTI-INSTANCE LOSS

Current MIL loss functions suffer from insufficient modeling of long-range dependencies and an optimization gap between instance-level and bag-level objectives, resulting in suboptimal hierarchical transitions. Therefore, a coupled multi-instance loss $\mathcal{L}_{\text{MIL}}$ is introduced, incorporating cross-instance interactions to model semantic dependencies and bridge the instance-bag optimization gap.

Conventional MIL losses are formulated as either bag-level ($\mathcal{L}_{\text{MIL}}$) or instance-level ($\mathcal{L}_{\text{INS}}$) objectives:

$$\mathcal{L}_{\text{MIL}} = CE\left(\frac{1}{N}\sum_{i=1}^{N} \lambda_i \cdot logits_i, y\right), \ \mathcal{L}_{INS} = \frac{1}{N}\sum_{i=1}^{N} \lambda_i \cdot CE\left(logits_i, y\right) \tag{4}$$

Here, $\mathcal{L}_{\text{MIL}}$ focuses on critical, high-evidence instances, while $\mathcal{L}_{\text{INS}}$ emphasizes individual accuracy but overlooks informative instance combinations. Both lack mechanisms for smooth hierarchical alignment and global-local interaction.

The proposed coupled multi-instance loss mitigates these problems by aggregating cross-instance interactions:

$$\mathcal{L}_{\text{CML}} = \frac{1}{2N^2} \sum_{i=1}^{N} \sum_{j=1}^{N} \lambda_i \cdot loss_j = \frac{1}{2N^2} \sum_{i=1}^{N} \lambda_i \cdot \sum_{j=1}^{N} loss_j \tag{5}$$

Here, both the attention weight $\lambda_i$ and the $loss_i$ derived from the non-independently optimized pathway (Optimizer1 in Fig. 4), where $loss_j$ denotes the cross-entropy between $logits_j$ and the pseudo labels. By weighting collective loss with total attention, this term promotes overall instance accuracy within relevant bags and effectively encourages instance representations to be discriminative. Additionally, an alternative coupled loss uses attention $\lambda_i'$ from the pathway optimized dependently (Optimizer2 in Fig. 4):

$$\mathcal{L}_{\text{CML}}' = \frac{1}{2N^2} \sum_{i=1}^{N} \sum_{j=1}^{N} \lambda_i' \cdot loss_j = \frac{1}{2N^2} \sum_{i=1}^{N} \lambda_i' \cdot \sum_{j=1}^{N} loss_j \tag{6}$$

The loss transfers richer global structure information, effectively guiding the joint optimization pathway toward both global consistency and local discrimination in instance features. The total loss,

$$L = \mathcal{L}_{\text{CML}} + \mathcal{L}_{\text{CML}}' + \mathcal{L}_{\text{MIL}} + \mathcal{L}_{\text{MIL}}' \tag{7}$$

prevents trivial solutions, synchronizes global and local objectives, and promotes hierarchical alignment and long-range semantic modeling. The formulation is compatible with the Hawkes pooling module to further enhance local associations and model robustness.

---

**Algorithm 1:** SuperMIL: Overall Architecture

**Input:** $D_{train} = \{(X_i, Y_i)\}_{i \in [1,N]}$, $D_{test} = \{X_i\}_{i \in [1,M]}$
**Output:** $pred\_labels$

1 initialization: Backbone $\psi_E(\theta_E)$, Hawkes Attention $\psi_H(\theta_H)$, $\psi_H'(\theta_H')$, Classifier $\psi_C$, $\psi_C'$, $optimizer(\theta_E, \theta_C \text{ and } \theta_H)$, $optimizer'(\theta_C' \text{ and } \theta_H')$, $pred\_labels = \{ \}$;

2 **while** *not converged* **do**    // Traing stage

3      $optimizer.zero\_grad()$, $optimizer'.zero\_grad()$;

4      Sample data $(X, Y)$ from $D_{train}$ ;    // $X \in \mathbb{R}^{W \times T}, Y \in \mathbb{R}^C$

5      $E = \psi_E(\theta_E, X)$ ;    // $E \in \mathbb{R}^{T \times D}$

6      $\Lambda, Logits = \psi_H\{\theta_H, E\}$ , $\psi_C\{\theta_C, E\}$;

7      $\Lambda', Logits' = \psi_H'\{\theta_H', E\}$ , $\psi_C'\{\theta_C', E\}$ ;    // $\Lambda \in \mathbb{R}^T, Logits \in \mathbb{R}^{T \times C}$

8      Compute the cross-entropy loss $\mathcal{L}_{MIL} + \mathcal{L}_{MIL}' + \mathcal{L}_{CML} + \mathcal{L}_{CML}'$ using $\Lambda$, $Logits$, $\Lambda'$, $Logits'$ and apply backpropagation ;    // $\mathcal{L}_{MIL} \in \mathbb{R}, \mathcal{L}_{CML} \in \mathbb{R}$

9      $optimizer.step()$, $optimizer'.step()$;

10 **for** $X$ *in* $D_{test}$ **do**    // Testing stage

11      $E = \psi_E(\theta_E, X)$ ;    // $E \in \mathbb{R}^{T \times D}$

12      $\Lambda, Logits = \psi_C'\{\theta_C', E\}$ ;    // $\Lambda \in \mathbb{R}^T, Logits \in \mathbb{R}^{T \times C}$

13      $\hat{Y} = argmax(mean(\Lambda \cdot P))$ ;    // $\hat{Y} \in \mathbb{R}$

14      $pred\_labels.append(\hat{Y})$;

15 **return** $pred\_labels$;

---

### 3.3 Pipeline of SuperMIL

To enhance clarity, the overall pipeline of SuperMIL is presented in this section, as shown in Algorithm 1. The training phase involves two iterative optimization pathways. The first jointly optimizes a feature extractor $\psi_E$, a classifier $\psi_C$, and a Hawkes attention module $\psi_H$ to learn discriminative instance features. The second optimizes a classifier $\psi_C'$ and a Hawkes attention module $\psi_H'$ to obtain bag-level predictions. In the testing phase, instance features are extracted by $\psi_E$ and passed to $\psi_C'$, and the final prediction is obtained by aggregating outputs from $\psi_C'$ in probability space using $\psi_H'$.

## 4 EXPERIMENTS

### 4.1 EXPERIMENTAL SETUP

**Dataset.** Evaluation is performed on 112 univariate datasets from the UCR archive Dauet & et al. (2019) and 26 multivariate datasets from the UEA archive Bagnall et al. (2018), encompassing a wide range of application domains.

**Metrics.** Classification accuracy, balanced accuracy (Bal. Accuracy), and area under the ROC curve (AUROC) serve as evaluation metrics. Comparative analysis employs average rankings and win/draw/loss counts; statistical significance is determined by the Wilcoxon signed-rank test (significance level 0.02).

**Implementation Details.** Experiments are conducted on a single NVIDIA GeForce RTX 3090 GPU (24GB VRAM) using PyTorch 2.0.0. All models are trained for 1500 epochs on standard train/test splits using the Adam optimizer (learning rate 0.001, no weight decay) with a batch size of 16.

**Baselines.** The TSC experiment involves deep learning models, including Fully Convolutional Network (FCN) Wang et al. (2017), Residual Network (ResNet) Wang et al. (2017), and InceptionTime (IT) Fawaz & et al. (2020), each evaluated with GAP, MIL, and SuperMIL frameworks. Non-deep learning algorithms include the feature-based Fresh Pipeline with Rotation Forest Classifier (FreshP) Middlehurst & Bagnall (2022), interval-based QUANTiles (Quant) Dempster et al. (2024), shapelet-based Random Dilated Shapelet Transformer (RDST) Guillaume et al. (2022), dictionary-based Word Extraction for Time Series Classification 2.0 (WEASEL2) Schäfer & Leser (2023), convolution-based Hydra combined with MultiRocket (Hydra-MR) Tan et al. (2022), and hybrid-based HIVE-COTE 2.0 (HC2) Middlehurst et al. (2021).

### 4.2 ABLATION EXPERIMENT

Table 1: The core components of the SuperMIL framework.

| Component | Description |
|---|---|
| SuperMIL framework | The framework iteratively optimizes instance-level pseudo-supervised learning and bag-level weak supervision. |
| Hawkes pooling module | It captures instance causality by decomposing excitations into directional similarity and instance dissimilarity. |
| Coupled multi-instance loss | It models inter-instance and collective loss-global attention interactions to align instance-bag objectives. |

The ablation study is designed to verify the effectiveness of each proposed module by incrementally integrating them into the MIL-IT baseline and evaluating their individual contributions, as shown in Table 1. To systematically assess the contribution of each component, IT is first retained as the backbone within the SuperMIL framework (Ablation2); subsequently, the multi-instance loss is incorporated (Ablation1); and finally, the Hawkes pooling layer is added to form the complete SuperMIL-IT model. The critical difference analysis presented in Fig. 6 demonstrates that each added component consistently enhances performance, thereby validating the effectiveness of the proposed modules. These findings highlight the necessity of optimizing both instance-level and bag-level discriminability, as improvements in instance representations directly contribute to enhanced bag-level classification. Furthermore, explicitly modeling inter-instance dependencies and causal structures is shown to be crucial for achieving high classification accuracy.

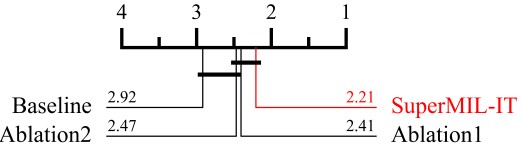

Figure 6: Critical difference diagram for the ablation study.

### 4.3 COMPARISON EXPERIMENTS

This section evaluates the performance of the proposed SuperMIL framework on TSC and multivariate TSC (MTSC) tasks, compared with MIL and GAP frameworks across various backbone models and under both single and ensemble settings.

**Generalization Experiment.** To investigate the generalizability of the SuperMIL framework, experiments are conducted on the UCR85 benchmark using multiple backbone models. As shown in Fig. 8 and Fig. 7, SuperMIL consistently improves the average performance of deep learning models, with its IT-based variant outperforming the hybrid method HC2 on UCR85. (HC2 is included only for reference as it is a meta-ensemble method and not directly comparable to single models. Results on UCR112 and UEA datasets are provided in Appendix C.)

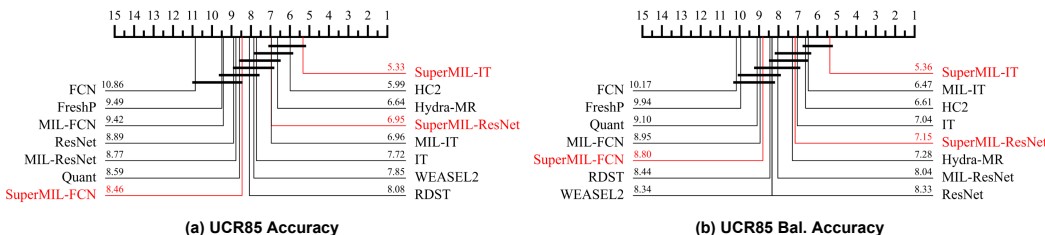

(a) UCR85 Accuracy          (b) UCR85 Bal. Accuracy

Figure 7: Critical difference diagram comparing SuperMIL with SOTAs.

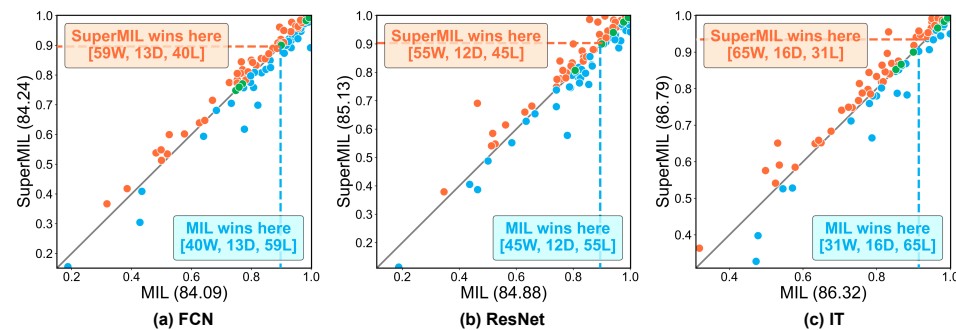

(a) FCN          (b) ResNet          (c) IT

Figure 8: Comparison of accuracy between SuperMIL (orange) and MIL (blue) on the 112 UCR datasets. Each point is a dataset, and points near the diagonal show similar performance. Dashed lines denote medians, and mean accuracies are provided for both methods across the various backbones.

**Perturbation Experiment.**

Fig. 9 depicts performance metric degradation as distinct explanation frameworks guide the progressive removal of time points. The line chart, with metric values on its left ordinate and corresponding improvements on its right, visualizes perturbation performance trends under attention-guided and random removal scenarios, while the bar chart directly compares the SuperMIL and MIL approaches. With the progressive removal of time points, SuperMIL exhibits marginally higher sensitivity than MIL, as the random removal of points disrupts the inter-dependencies between instances, which impacts SuperMIL more than MIL. In contrast, SuperMIL demonstrates greater stability when time point removal is guided by instance importance. This resilience is attributed to the more discriminative representation acquired by each instance, which allows for correct classification based on the remaining instances, even if the most critical ones are elided.

**Efficiency Experiment.** This experiment evaluates the training efficiency of SuperMIL in comparison to GAP and MIL. As shown in Fig. 10, SuperMIL achieves the accuracy that GAP and MIL reach after 1,500 epochs in only about 350 epochs on average. Although each epoch of SuperMIL requires more computation, the total time to reach comparable accuracy is still reduced. Furthermore, SuperMIL continues to improve with additional training, indicating greater potential for further performance gains.

## 5 CONCLUSION

The proposed SuperMIL overcomes the limitation of insufficient instance-level feature learning by iteratively decoupling optimizes instance-level pseudo-supervision and bag-level weak supervision

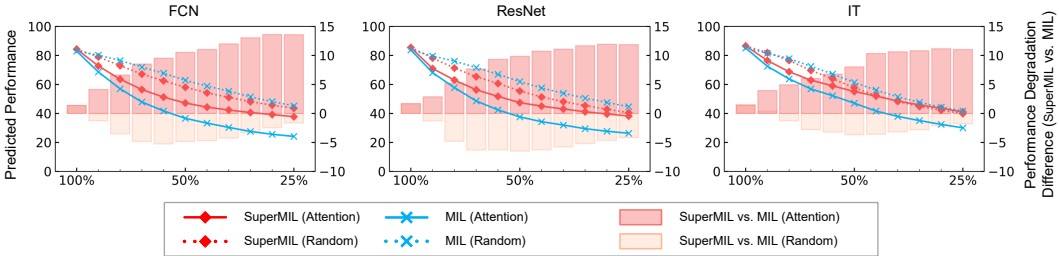

Figure 9: Perturbation curves showing the rate at which the model prediction decays when time points are removed following the orderings proposed by the different interpretability methods.

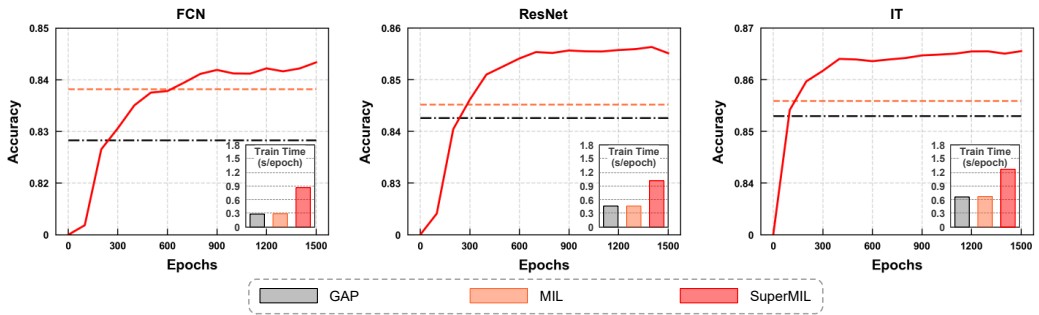

Figure 10: Performance trends with increasing training epochs for SuperMIL, MIL, and GAP. MIL and GAP are shown at 1500 epochs; SuperMIL reaches their accuracy earlier despite slower per-epoch speed, offering faster convergence to target accuracy and a higher performance ceiling.

Table 2: Performance on 85 UCR datasets under ensemble learning in form: mean / rank / wins.

| Method | Accuracy | | | Bal.Accuracy | | | AUROC | | |
|---|---|---|---|---|---|---|---|---|---|
| | Mean | Rank | Wins | Mean | Rank | Wins | Mean | Rank | Wins |
| Hydra-MR | 85.65 | 6.64 | 17 | 83.07 | 7.28 | 10 | 88.79 | 12.39 | 8 |
| FCN | 82.83 | 10.86 | 7 | 80.39 | 10.17 | 8 | 92.92 | 9.22 | 8 |
| MIL-FCN | 83.82 | 9.42 | 5 | 81.36 | 8.95 | 4 | 93.38 | 7.76 | 9 |
| SuperMIL-FCN | 84.34 | 8.46 | 7 | 81.32 | 8.80 | 7 | 93.36 | 7.46 | 7 |
| ResNet | 84.26 | 8.89 | 7 | 81.91 | 8.33 | 7 | 93.68 | 7.31 | 10 |
| MIL-ResNet | 84.52 | 8.77 | 8 | 82.25 | 8.04 | 9 | 93.86 | 6.86 | 9 |
| SuperMIL-ResNet | 85.51 | 6.95 | 14 | 82.78 | 7.15 | 15 | 93.84 | 6.29 | 15 |
| IT | 85.30 | 7.72 | 13 | 83.24 | 7.04 | 15 | 93.93 | 6.70 | 15 |
| MIL-IT | 85.59 | 6.96 | 11 | 83.40 | 6.47 | 16 | 93.90 | 6.52 | __18__ |
| SuperMIL-IT | __86.72__ | __5.33__ | __23__ | __84.37__ | __5.36__ | __20__ | __94.23__ | __5.66__ | 17 |
| *HC2* | *86.07* | *5.99* | *20* | *83.13* | *6.61* | *17* | *95.03* | *4.33* | *35* |

learning. Moreover, SuperMIL integrates a Hawkes pooling module and a coupled multi-instance loss, synergizing the capture of local causal causality and global instance semantics. The former captures local causality by decomposing inter-instance excitations into directional similarity and instance differences, modeling both correlation and evolutionary dynamics, enabling effective temporal aggregation and enhancing robustness for complex TSC tasks. The latter synchronizes global and local objectives by modeling inter-instance and collective loss-global attention interactions, promoting hierarchical alignment and long-range semantic modeling. Comprehensive experiments on UCR and UEA benchmarks demonstrate that SuperMIL outperforms MIL and GAP frameworks regarding accuracy, interpretability, and robustness, particularly on complex datasets. These findings highlight the advantages of joint local-global optimization and temporal dependency modeling. However, a current limitation is its primary application to time series data, as domains with other types of semantic relationships have not yet been explored. Future work will, therefore, extend validation to these broader application domains.

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

## A    APPENDIX

The following discussion explores the advantages of SuperMIL, tracing them to its hierarchical modeling approach:

- **Feature Extraction Layer:** This layer extracts discriminative instance features within the feature space, improving instance feature representation.
- **Hawkes Pooling Layer:** This layer models dynamic causal interdependencies among instances, achieving causal aggregation in the probability space.
- **Loss Function:** This layer enforces semantic consistency between instance and bag representations, enhancing the modelling of long-range and short-range dependencies.

The appendix is structured as follows: Section B states the use of LLMs, and C discusses the statistical findings from extensive experiments. Section D outlines the detailed network architecture of SuperMIL along with its time and space complexity. Section E presents the detailed experimental results for each dataset.

## B    USE OF LLMS

In compliance with ICLR 2026 policies on LLM usage, we disclose that LLMs were only used for language polishing of this manuscript, specifically to improve grammatical accuracy, text fluency, and wording clarity. No LLM was used for core research tasks (e.g., ideation, experiment design, code generation, data analysis, or drafting key sections). The authors bear full responsibility for the manuscript's accuracy and integrity, and all LLM-polished content has been reviewed and validated.

## C    EXPERIMENTAL EVALUATION

This section reports experimental results for SuperMIL across UCR and UEA[1] datasets, including an analysis of performance relative to dataset attributes. Extended UCR results can be found in Sections C.1 and C.2, and UEA results can be found in Section C.3. The interplay between SuperMIL performance and dataset properties is further explored in Section C.4, with interpretability analysis detailed in Section C.5. Section C.6 provides a systematic analysis of hyperparameter sensitivity. Notably, the backbone architectures in these experiments (including FCN, ResNet, IT, ConvTran, and TodyNet) are selected for the successful reproduction of accuracies reported in their original publications. Architectures not meeting this validation standard are therefore not included in this study.

### C.1    UCR85 RESULTS

Table A.1 compares SuperMIL with state-of-the-art algorithms on 85 UCR datasets, where the hybrid method, HC2, is included for reference only and is not directly compared.

The radar chart in Fig. A.1 (left) further illustrates the significant enhancements brought by SuperMIL in the TSC task. These improvements are observed consistently across various backbones, surpassing both MIL and GAP frameworks, and detailed results are summarized in Table 2.

To examine the relationship between the effectiveness of SuperMIL and data difficulty, datasets are stratified by classification difficulty (criteria and further analyses in Appendix B). As shown in Fig. A.1 (right), SuperMIL delivers the most substantial improvements on challenging datasets, showcasing its capability to enhance model robustness in complex scenarios.

### C.2    UCR112 RESULTS

Table A.2 compares SuperMIL with state-of-the-art algorithms on 112 UCR datasets, where the hybrid method, HC2, is included for reference only and is not directly compared. Fig. A.2 presents

---

[1]https://timeseriesclassification.com/

Table A.1: Performance on 85 UCR datasets under ensemble learning in form: mean / rank / wins.

| Method | Accuracy | | | Bal.Accuracy | | | AUROC | | |
|---|---|---|---|---|---|---|---|---|---|
| | Mean | Rank | Wins | Mean | Rank | Wins | Mean | Rank | Wins |
| FreshP | 83.29 | 9.49 | 12 | 80.07 | 9.94 | 12 | 94.38 | 7.14 | 15 |
| Quant | 84.44 | 8.59 | 7 | 81.40 | 9.10 | 5 | **94.46** | 6.35 | 11 |
| RDST | 84.99 | 8.08 | 8 | 82.19 | 8.44 | 7 | 88.22 | 12.91 | 6 |
| WEASEL2 | 84.96 | 7.85 | 11 | 82.25 | 8.34 | 10 | 88.32 | 13.10 | 6 |
| Hydra-MR | 85.65 | 6.64 | 17 | 83.07 | 7.28 | 10 | 88.79 | 12.39 | 8 |
| FCN | 82.83 | 10.86 | 7 | 80.39 | 10.17 | 8 | 92.92 | 9.22 | 8 |
| MIL-FCN | 83.82 | 9.42 | 5 | 81.36 | 8.95 | 4 | 93.38 | 7.76 | 9 |
| SuperMIL-FCN | 84.34 | 8.46 | 7 | 81.32 | 8.80 | 7 | 93.36 | 7.46 | 7 |
| ResNet | 84.26 | 8.89 | 7 | 81.91 | 8.33 | 7 | 93.68 | 7.31 | 10 |
| MIL-ResNet | 84.52 | 8.77 | 8 | 82.25 | 8.04 | 9 | 93.86 | 6.86 | 9 |
| SuperMIL-ResNet | 85.51 | 6.95 | 14 | 82.78 | 7.15 | 15 | 93.84 | 6.29 | 15 |
| IT | 85.30 | 7.72 | 13 | 83.24 | 7.04 | 15 | 93.93 | 6.70 | 15 |
| MIL-IT | 85.59 | 6.96 | 11 | 83.40 | 6.47 | 16 | 93.90 | 6.52 | **18** |
| SuperMIL-IT | **86.72** | **5.33** | **23** | **84.37** | **5.36** | **20** | 94.23 | **5.66** | 17 |
| *HC2* | *86.07* | *5.99* | *20* | *83.13* | *6.61* | *17* | *95.03* | *4.33* | *35* |

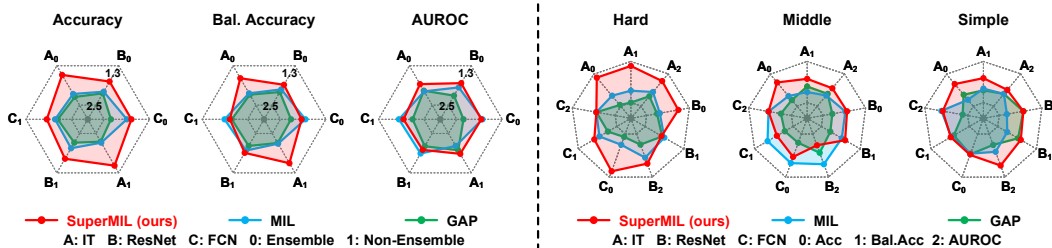

Figure A.1: Radar charts comparing classification performance of SuperMIL, MIL, and GAP across various backbones (left), and illustrating SuperMIL improvements on simple, middle, and hard datasets across different backbones (right) in the TSC task.

the results of the Wilcoxon signed-rank test conducted across all classifiers, indicating that SuperMIL-based approaches outperformed GAP-based and MIL-based methods. Furthermore, scatter plots offer an intuitive comparison of different backbones in the SuperMIL and MIL frameworks, as illustrated in Fig. A.3. These plots highlight that various backbones, when embedded into the SuperMIL framework, achieve a greater number of wins or draws, with records of 62/112 (55.36%), 67/112 (59.82%), and 81/112 (72.32%) respectively.

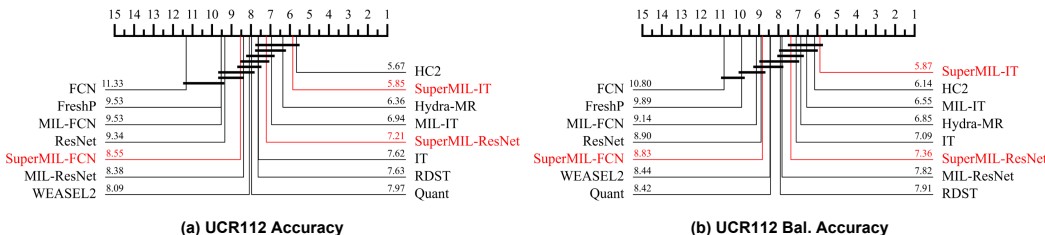

Figure A.2: Critical difference diagram comparing SuperMIL with SOTAs on 112 UCR datasets.

## C.3 UEA26 RESULTS

Table A.3 compares SuperMIL with MIL and GAP on 26 UEA datasets, and the Wilcoxon signed-rank tests confirm the performance of SuperMIL over GAP and MIL in MTSC tasks, as shown in Fig. A.4. The radar chart in Fig. A.5 offers an intuitive visual comparison of SuperMIL's comprehensive

Table A.2: Performance on 112 UCR datasets under ensemble learning in form: mean / rank / wins.

| Method | Accuracy | | | Bal.Accuracy | | | AUROC | | |
|---|---|---|---|---|---|---|---|---|---|
| | Mean | Rank | Wins | Mean | Rank | Wins | Mean | Rank | Wins |
| FreshP | 84.16 | 9.53 | 19 | 81.71 | 9.89 | 19 | 95.09 | 7.28 | 19 |
| Quant | 85.44 | 7.79 | 17 | 83.06 | 8.42 | 15 | **95.35** | 6.20 | 20 |
| RDST | 86.31 | 7.63 | 14 | 84.20 | 7.91 | 13 | 89.67 | 12.33 | 21 |
| WEASEL2 | 86.05 | 8.09 | 12 | 84.02 | 8.44 | 11 | 89.53 | 12.99 | 7 |
| Hydra-MR | **86.80** | 6.36 | 25 | 84.87 | 6.85 | 24 | 90.13 | 11.80 | 14 |
| FCN | 81.89 | 11.33 | 7 | 79.90 | 10.80 | 8 | 92.86 | 9.62 | 10 |
| MIL-FCN | 84.09 | 9.53 | 6 | 82.24 | 9.14 | 5 | 93.96 | 7.82 | 14 |
| SuperMIL-FCN | 84.24 | 8.55 | 11 | 81.90 | 8.83 | 11 | 93.86 | 7.30 | 13 |
| ResNet | 83.62 | 9.34 | 8 | 81.75 | 8.90 | 8 | 93.81 | 7.90 | 12 |
| MIL-ResNet | 84.88 | 8.38 | 11 | 83.18 | 7.82 | 12 | 94.41 | 6.85 | 16 |
| SuperMIL-ResNet | 85.12 | 7.21 | 18 | 83.01 | 7.36 | 19 | 94.16 | 6.48 | 20 |
| IT | 86.01 | 7.62 | 16 | 84.35 | 7.09 | 18 | 94.63 | 6.77 | 18 |
| MIL-IT | 86.32 | 6.94 | 25 | 84.62 | 6.55 | 20 | 94.65 | 6.51 | 21 |
| SuperMIL-IT | 86.79 | **5.85** | **29** | **84.94** | **5.87** | **26** | 94.68 | **5.84** | **27** |
| *HC2* | *87.64* | *5.67* | *32* | *85.43* | *6.14* | *29* | *95.03* | *4.31* | *51* |

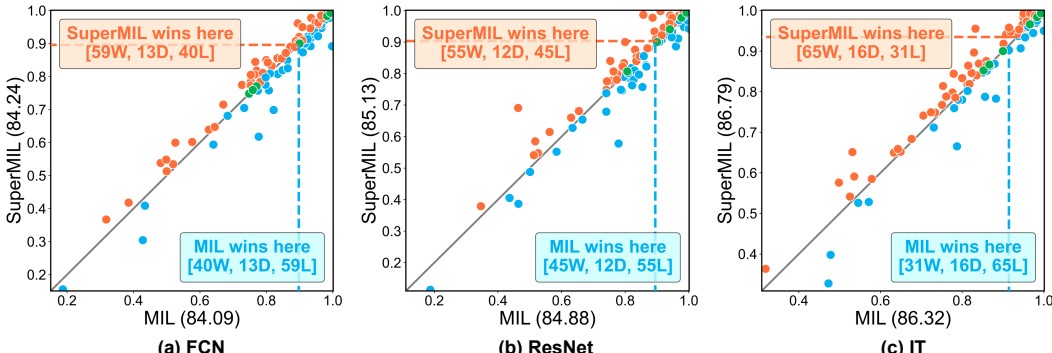

Figure A.3: Comparison of accuracy between SuperMIL (orange) and MIL (blue) on the 112 UCR datasets. Each point is a dataset, and points near the diagonal show similar performance. Dashed lines denote medians, and mean accuracies are provided for both methods across the various backbones.

strengths: its largest coverage area signifies robust performance across diverse backbones and multiple evaluation metrics under ensemble and non-ensemble configurations.

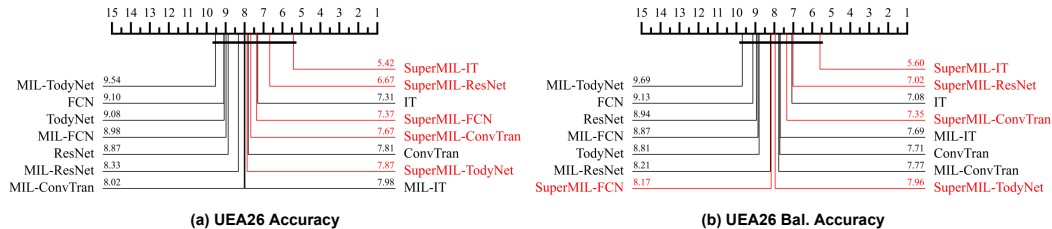

Figure A.4: Critical difference diagram comparing SuperMIL with SOTAs on 26 UEA datasets.

## C.4 PERFORMANCE OF DATASET PROPERTIES

As indicated in Fig. A.1 (right), SuperMIL achieves substantial performance gains, particularly on challenging datasets. To explore the dependency of these improvements on dataset properties, the

Table A.3: Performance on 26 UEA datasets under ensemble learning in form: mean / rank / wins.

| Method | Accuracy | | | Bal.Accuracy | | | AUROC | | |
|---|---|---|---|---|---|---|---|---|---|
| | Mean | Rank | Wins | Mean | Rank | Wins | Mean | Rank | Wins |
| FCN | 70.57 | 9.10 | 2 | 69.71 | 9.13 | 2 | 82.51 | 8.83 | 2 |
| MIL-FCN | 70.83 | 8.98 | 1 | 69.94 | 8.87 | 1 | 83.01 | 8.13 | 2 |
| SuperMIL-FCN | 70.99 | 7.37 | 4 | 69.69 | 8.17 | 4 | 83.10 | 7.52 | 4 |
| ResNet | 70.96 | 8.87 | 1 | 70.00 | 8.94 | 1 | 82.56 | 9.40 | 2 |
| MIL-ResNet | 71.12 | 8.33 | 2 | 70.15 | 8.21 | 2 | 82.83 | 8.90 | 2 |
| SuperMIL-ResNet | 71.20 | 7.02 | **8** | 70.07 | 7.02 | 6 | 83.06 | 7.02 | **7** |
| IT | 70.20 | 7.31 | 2 | 71.26 | 7.08 | 3 | 83.66 | 7.46 | 3 |
| MIL-IT | 71.77 | 7.98 | 3 | 70.80 | 7.69 | 3 | 83.41 | 8.15 | 4 |
| SuperMIL-IT | **72.58** | **5.42** | 7 | **71.46** | **5.60** | **7** | 83.17 | 7.38 | 6 |
| ConvTran | 71.52 | 7.81 | 2 | 70.51 | 7.71 | 2 | 83.93 | 6.88 | 2 |
| MIL-ConvTran | 71.81 | 8.02 | 2 | 70.76 | 7.77 | 2 | 84.16 | **6.25** | 3 |
| SuperMIL-ConvTran | 72.08 | 7.67 | 2 | 71.13 | 7.35 | 2 | 83.85 | 7.17 | 4 |
| TodyNet | 71.87 | 9.08 | 1 | 70.88 | 8.81 | 2 | 84.35 | 9.06 | 3 |
| MIL-TodyNet | 71.16 | 9.54 | 2 | 70.00 | 9.69 | 2 | 83.82 | 9.63 | 3 |
| SuperMIL-TodyNet | 72.29 | 7.67 | 5 | 71.06 | 7.96 | 5 | **84.42** | 8.19 | 5 |

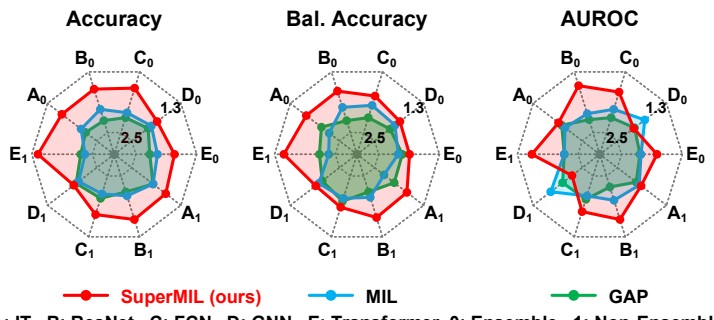

Figure A.5: Radar charts comparing classification performance of SuperMIL, MIL, and GAP across various backbones on 26 UEA datasets.

results of the 112 UCR datasets are stratified based on several key properties: the length of time series, the number of time series, the number of classes, and the class imbalance.

Fig. A.6 illustrates that SuperMIL achieves superior performance on datasets characterized by longer time series, a larger volume of samples, and a greater number of classes, aligning with the findings presented in Fig. 8. MIL struggles with such datasets due to its dependence on sparse instances, which leads to insufficient information utilization. In contrast, SuperMIL optimizes at the instance level, which simulates data augmentation, enhancing information exploitation and generalization.

To evaluate the impact of varying degrees of class imbalance on SuperMIL, dataset imbalance is first quantified using normalized Shannon entropy.

$$\text{Dataset Balance} = -\frac{1}{\log c} \sum_{i=1}^{c} \frac{c_i}{n} \log\left(\frac{c_i}{n}\right) \quad \text{(A.1)}$$

where $c$ denotes the number of classes, $c_i$ is the number of samples in the $i$-th class, and $n$ represents the total number of samples. Datasets are subsequently partitioned based on entropy thresholds of 0.9 and 0.99. An entropy range of 0 to 0.9 signifies severe imbalance, 0.9 to 0.99 indicates moderate imbalance, and 0.99 to 1.0 represents mild imbalance or perfect balance. Fig. A.7 illustrates the comparative performance of SuperMIL against MIL across class imbalance, showing that SuperMIL consistently outperforms MIL on severely imbalanced datasets in various backbone architectures. This observation suggests a notable advantage of SuperMIL in mitigating class imbalance, aligning with its inherent data generalization capabilities discussed earlier.

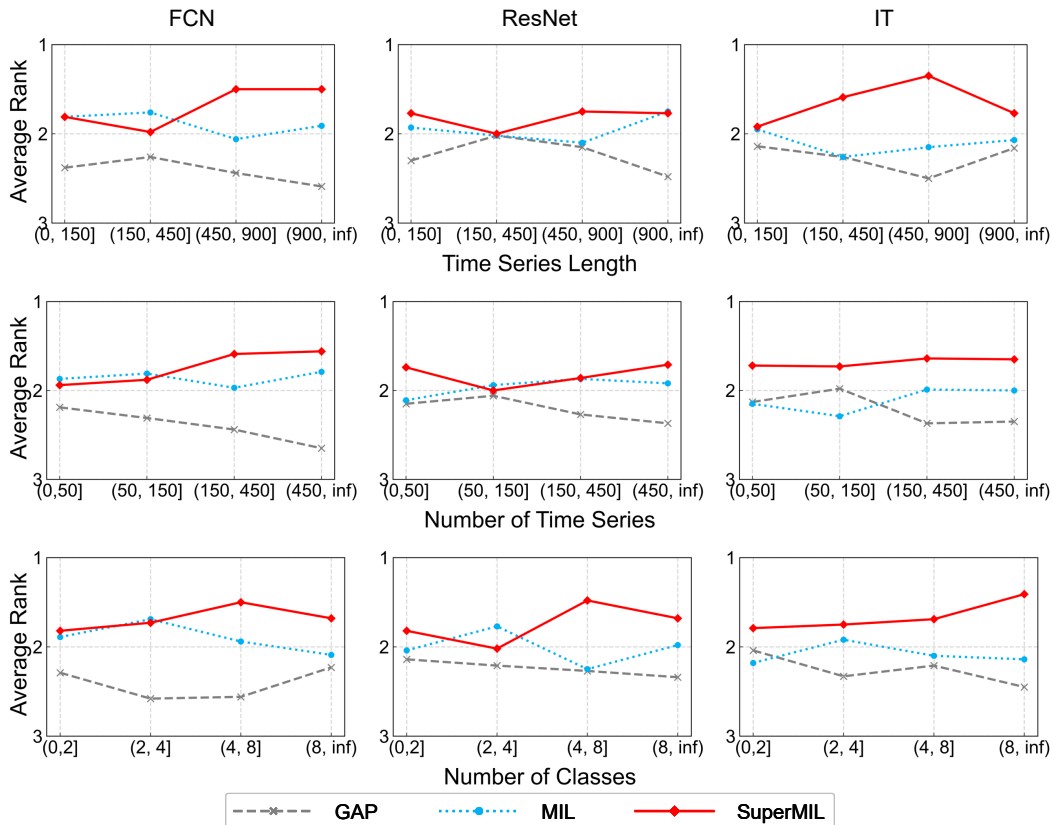

Figure A.6: Comparison of SuperMIL, MIL, and GAP regarding the impact of time series length, the number of time series, and the number of classes, as measured by average accuracy rankings on 112 UCR datasets.

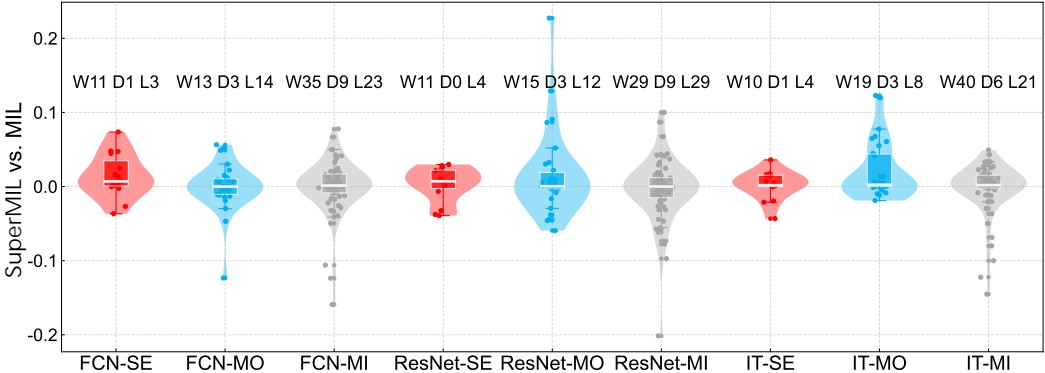

Figure A.7: Impact of data imbalance on SuperMIL versus MIL performance, evaluated by accuracy differentials across 112 UCR datasets. W / D / L indicates the number of datasets where SuperMIL wins, draws, or loses against MIL.

## C.5 INTERPRETABILITY AND VISUALIZATION EXPERIMENTS

Current MIL methods enhance interpretability through discriminative instance sparsity without ensuring instance correctness. Inaccurate instance predictions severely weaken interpretability, leading to misleading explanations that deviate from discriminative features. However, SuperMIL prioritizes accurate and stable instance-level predictions, which are foundations for reliable explanations and robust bag-level classification. Fig. A.8 provides visualizations derived from multiple datasets to

facilitate a more intuitive comprehension of the effects of SuperMIL on instance features. These demonstrate that SuperMIL-based methods substantially augment instance-level prediction accuracy, contributing to an elevation in bag-level precision.

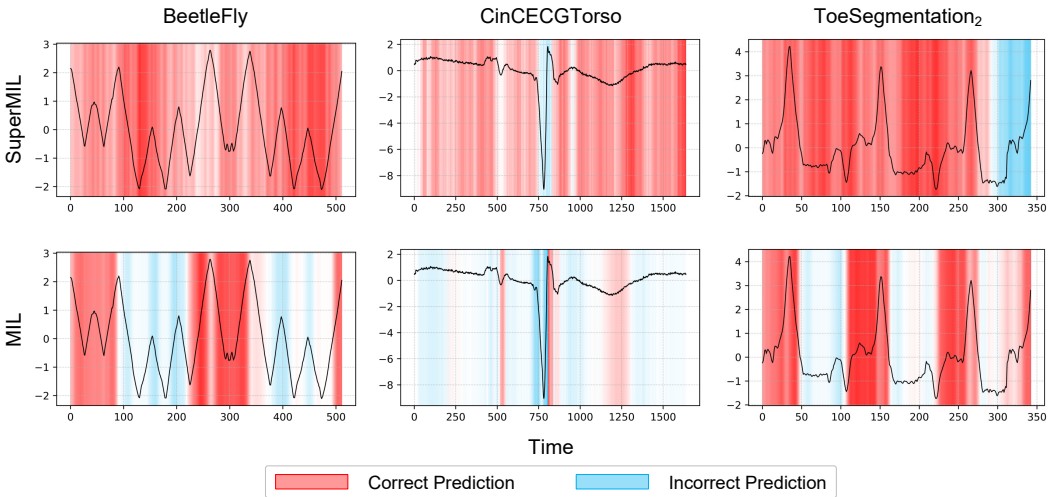

Figure A.8: Visual comparison of SuperMIL and MIL on UCR datasets. Red denotes correctly predicted instances, while blue indicates incorrectly predicted ones. The intensity of the colour represents the predictive probability, with darker shades signifying higher probabilities.

### C.6 HYPERPARAMETER SENSITIVITY EXPERIMENT

SuperMIL uses a granularity parameter, $g$ (default $g = 50$), to balance computational efficiency and representation power by compressing embeddings into $g$ instances. Its robustness was tested across 10% to 100% of the total UCR time points. As shown in Fig. A.9 and Table A.4, key evaluation metrics were stable across settings, indicating low sensitivity to performance variations. The analysis included the datasets: BeetleFly, CinCECGTorso, Ham, InlineSkate, Phoneme, ToeSegmentation$_2$, and WormsTwoClass.

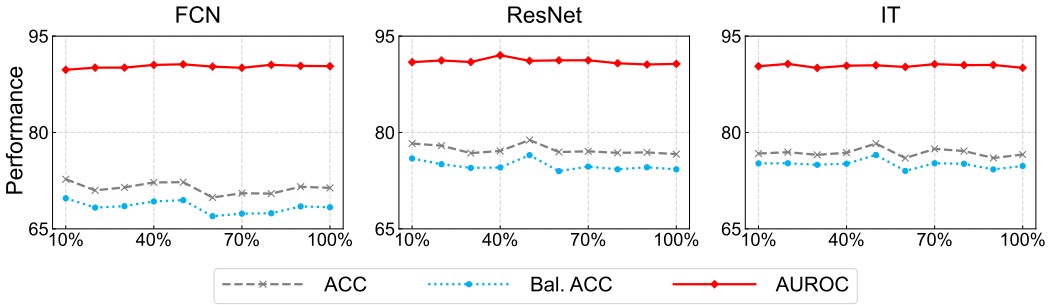

Figure A.9: Sensitivity analysis of granularity parameter. Evaluation as $g$ varies from 10% to 100% of the original $t$ time points.

Table A.4: Sensitivity analysis of hyperparameter parameters. The mean accuracy and standard deviation under different hyperparameter settings are presented.

| Metric | Accuracy | | | Bal.Accuracy | | | AUROC | | |
|---|---|---|---|---|---|---|---|---|---|
| | FCN | ResNet | IT | FCN | ResNet | IT | FCN | ResNet | IT |
| Mean | 71.35 | 77.34 | 76.85 | 68.40 | 74.83 | 75.04 | 90.26 | 91.08 | 90.38 |
| Standard Deviation | 0.85 | 0.71 | 0.63 | 0.89 | 0.75 | 0.63 | 0.25 | 0.39 | 0.21 |

# D  IMPLEMENTATION DETAILS

This section details the network architecture of Hawkes pooling and analyzes its time complexity and parameter count.

## D.1  DETAILED NETWORK DESIGN

Table A.5 details the architecture of the Hawkes pooling method. Each row describes a unique component within this pooling architecture that processes different input tensors, and the variables adhere to the definitions outlined in Section 3. Additionally, $b$ denotes the batch size, $t$ represents the time series length, and $c$ is the number of classes. The constant value of 128 in the table represents the feature dimension size, denoted as $d$.

Table A.5: Network architecture of the Hawkes pooling method.

| Process | Layer | Input | Output |
|---|---|---|---|
| Direction Similarity ($\psi_A$) | Linear + tanh | $b \times t \times 128 \ (e_i)$ | $b \times t \times 128 \ (\psi_A (e_i))$ |
| Instance Difference ($\psi_D$) | Linear + tanh 
 Linear + sigmoid | $b \times t \times 128 \ (e_j - e_i)$ 
 $b \times t \times 128$ | $b \times t \times 8$ 
 $b \times t \times 1 \ (d_{ji})$ |
| Self-Importance ($\psi_S$) | Linear + tanh 
 Linear + sigmoid | $b \times t \times 128 \ (e_i)$ 
 $b \times t \times 128$ | $b \times t \times 8$ 
 $b \times t \times 1 \ (s_i)$ |
| Classifier ($\psi_C$) | Linear | $b \times t \times 128 \ (e_i)$ | $b \times t \times c \ (logits_i)$ |
| Hawkes Pooling | Attn. Weighting 
 Mean | $b \times t \times c \ (logits_i$ and $\lambda_i)$ 
 $b \times t \times c$ | $b \times t \times c$ 
 $b \times 1 \times c \ (\overline{logits})$ |

## D.2  RUN TIME ANALYSIS

An examination of the time complexity of SuperMIL, MIL, and GAP is presented after the detailed structural analysis of Hawkes pooling. Because of a common backbone network, these methods' runtime discrepancies are primarily attributable to their respective pooling layers. Therefore, this section compares the Hawkes pooling against GAP and MIL. The time complexities for GAP and MIL are derived from MILLET. For consistency with this approach, the calculation of Hawkes pooling runtime omits activation functions and non-linear operations, retaining only the actual multiplication operations for a focused comparison. The complexity analysis is presented in Table A.6, and Fig. A.10 further illustrates the growth in time complexity as the number of class and time series lengths increase.

Table A.6: Time complexity analysis (number of real multiplications) of different pooling approaches. Furthermore, the scaling behavior of these multiplication counts concerning the number of timesteps ($t$) and the number of classes ($c$) is also detailed.

| Method | Number of Real Multiplications | Scale w.r.t $t$ | Scale w.r.t $c$ |
|---|---|---|---|
| GAP | $d \times t \times c + d \times t + d \times c$ | $d \times c + d$ | $d \times t + d$ |
| MIL | $d \times t \times c + d \times t \times a + (a + 2c) \times t$ | $d \times c + d \times a + a + 2c$ | $d \times t + 2t$ |
| Hawkes | $d \times t \times c + (a + 2c) \times t$ 
 $+ d \times t \times k + (d \times a + a) \times t \times k + t$ | $d \times c + a + 2c$ 
 $+ d \times k + (d \times a + a) \times k + 1$ | $d \times t + 2t$ |

## D.3  NUMBER OF PARAMETERS

The model parameters for GAP, MIL, and SuperMIL are provided in Table A.7, and the SuperMIL implementation shows a modest parameter increase of at most 4.6% from the GAP model.

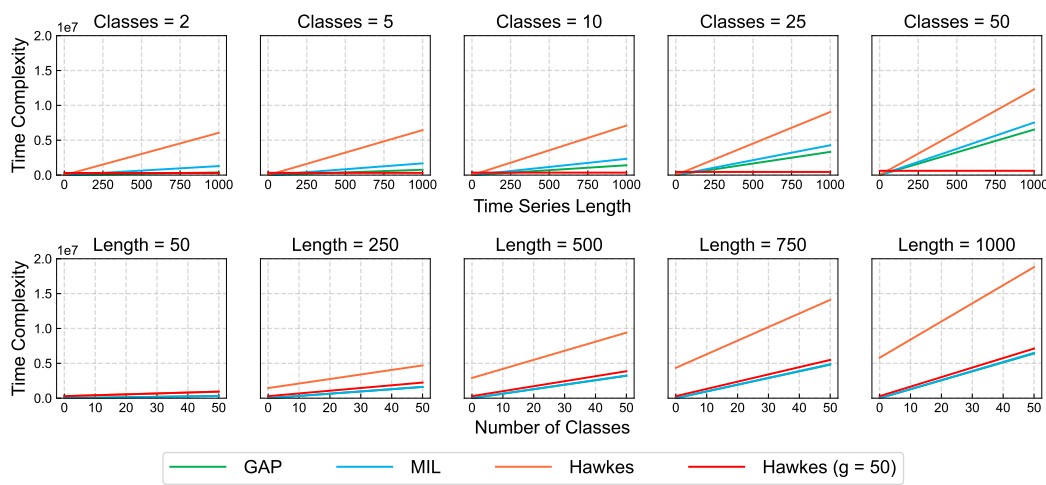

Figure A.10: Time complexity (measured by real multiplications) for the pooling methods employed. Top: Scalability with time series length, shown for varying numbers of classes. Bottom: Scalability with the number of classes, shown for varying time series lengths.

Table A.7: Number of model parameters for Fish. Percentages indicate the relative increase in parameters from GAP model.

| Pooling | FCN | ResNet | IT |
|---|---|---|---|
| GAP | 265.6K | 504.9K | 422.3K |
| MIL | 266.6K (+0.4%) | 505.9K (+0.2%) | 423.3K (+0.2%) |
| SuperMIL | 277.8K (+4.6%) | 517.1K (+2.4%) | 434.5K (+2.9%) |

## E  ADDITIONAL RESULTS

Tables A.8 and A.9 present the average accuracies of SuperMIL on the 112 UCR and 26 UEA datasets, respectively. Comprehensive results are detailed further at our codebase[2].

Table A.8: Classification accuracy on 112 UCR datasets.

| DataSets | GAP -FCN | MIL -FCN | SuperMIL -FCN | GAP -ResNet | MIL -ResNet | SuperMIL -ResNet | GAP -IT | MIL -IT | SuperMIL -IT |
|---|---|---|---|---|---|---|---|---|---|
| Adiac | 84.91 | 87.21 | **89.00** | 81.33 | 83.12 | 87.55 | 83.89 | 85.17 | 84.74 |
| ArrowHead | 85.71 | 86.29 | 84.95 | 87.43 | 87.43 | **88.00** | 85.14 | 86.29 | 86.86 |
| Beef | 73.33 | 76.67 | 84.44 | 80.00 | 80.00 | **90.00** | 90.00 | 86.67 | 86.67 |
| BeetleFly | 85.00 | 80.00 | 85.00 | 90.00 | 80.00 | 90.00 | 85.00 | 95.00 | **98.33** |
| BirdChicken | **100.00** | 90.00 | 90.00 | 90.00 | 90.00 | 90.00 | 100.00 | 95.00 | 98.33 |
| Car | 88.33 | 93.33 | 91.67 | 91.67 | **95.00** | 93.33 | 91.67 | 93.33 | 94.44 |
| CBF | 99.22 | 99.67 | 98.59 | 99.78 | 99.56 | 99.63 | **100.00** | 100.00 | 99.96 |
| ChlorineC | 83.15 | 83.44 | 80.76 | 86.41 | 86.54 | 82.61 | **88.18** | 86.74 | 86.83 |
| CinCECGT | 83.77 | 92.61 | 92.15 | 80.94 | 88.62 | **97.71** | 83.41 | 83.26 | 95.53 |
| Coffee | 100.00 | 100.00 | 100.00 | 100.00 | 100.00 | 100.00 | 100.00 | 100.00 | 100.00 |
| Computers | 82.00 | 81.60 | 83.20 | 82.00 | 83.60 | **85.20** | 79.20 | 80.00 | 78.00 |
| CricketX | 78.46 | 77.44 | 75.73 | 80.77 | 81.28 | 78.63 | 85.13 | 85.13 | **85.30** |
| CricketY | 78.97 | 80.77 | 75.81 | 82.56 | 82.31 | 79.57 | 86.67 | 84.10 | **86.92** |
| CricketZ | 79.74 | 79.74 | 77.52 | 81.79 | 82.31 | 80.09 | 86.92 | **87.95** | 86.84 |
| DiatomSizeR | 95.10 | 95.10 | 94.66 | 96.41 | 94.77 | 93.14 | 96.08 | **98.37** | 98.37 |
| DistalPOAG | 69.78 | 72.66 | 77.46 | 73.38 | 75.54 | **77.70** | 71.94 | 70.50 | 74.10 |
| DistalPOC | 76.81 | 79.35 | 80.07 | 80.43 | **82.61** | 78.02 | 76.45 | 78.99 | 78.02 |
| DistalPTW | **69.06** | 68.35 | 68.11 | 66.91 | 65.47 | 68.11 | 66.91 | 67.63 | 68.35 |
| Earthquakes | 72.66 | **74.82** | 74.82 | 72.66 | 74.10 | **74.82** | 74.10 | 73.38 | **74.82** |
| ECG200 | 88.00 | **91.00** | 87.33 | 88.00 | 87.00 | 88.00 | **91.00** | 90.00 | 90.00 |
| ECG5000 | 94.02 | 94.09 | 94.27 | 94.02 | 93.98 | **94.69** | 93.82 | 94.58 | 94.56 |
| ECGFiveDays | 98.95 | 99.19 | 97.87 | 99.65 | 99.42 | 99.23 | **100.00** | 100.00 | 100.00 |
| ElectricDevices | 72.75 | 75.00 | 74.73 | **75.05** | 74.09 | 73.77 | 73.92 | 73.05 | 71.16 |

---

[2]https://anonymous.4open.science/r/SuperMIL

Table A.8: Classification accuracy on 112 UCR datasets.

| DataSets | GAP-FCN | MIL-FCN | SuperMIL-FCN | GAP-ResNet | MIL-ResNet | SuperMIL-ResNet | GAP-IT | MIL-IT | SuperMIL-IT |
|---|---|---|---|---|---|---|---|---|---|
| FaceAll | 94.44 | 93.31 | 89.19 | 82.66 | **95.98** | 88.60 | 81.42 | 95.62 | 93.55 |
| FaceFour | 93.18 | **96.59** | 94.70 | 95.45 | 95.45 | 95.83 | 94.32 | **96.59** | **96.59** |
| FacesUCR | 94.44 | 94.54 | 93.09 | 95.12 | 96.34 | 96.11 | 97.07 | 96.59 | **97.11** |
| FiftyWords | 66.37 | 67.03 | 71.50 | 77.36 | 78.68 | 79.34 | 84.84 | 85.93 | **87.55** |
| Fish | 95.43 | 97.71 | 97.90 | 98.29 | 97.71 | 97.71 | 97.14 | 98.29 | **98.48** |
| FordA | 91.97 | 93.48 | 97.73 | 94.24 | 94.17 | **98.41** | 95.91 | 96.14 | 96.19 |
| FordB | 77.90 | 78.52 | 79.67 | 82.47 | 81.98 | 81.73 | 85.93 | 86.05 | **86.46** |
| GunPoint | **100.00** | 98.00 | **100.00** | **100.00** | 99.33 | **100.00** | **100.00** | 98.67 | 99.78 |
| Ham | 71.43 | 73.33 | 70.48 | 77.14 | **79.05** | 74.60 | 73.33 | 72.38 | 74.92 |
| HandOutlines | 91.89 | 93.78 | 92.79 | 92.97 | 94.05 | 94.05 | 95.95 | 95.68 | **96.31** |
| Haptics | 49.03 | 51.95 | 53.46 | 52.92 | 52.60 | 54.76 | 52.92 | 53.57 | **59.09** |
| Herring | 64.06 | 64.06 | 59.38 | 64.06 | 56.25 | 61.46 | 50.00 | 53.12 | **65.10** |
| InlineSkate | 45.45 | 48.18 | 53.82 | 42.73 | 46.36 | **69.09** | 50.18 | 49.82 | 57.58 |
| InsectWS | 40.91 | 38.54 | 41.80 | 53.03 | 50.10 | 48.77 | 63.64 | 63.28 | **64.97** |
| ItalyPowerD | 96.21 | 96.11 | 95.92 | 96.31 | 94.56 | 96.44 | 96.50 | 95.43 | **96.63** |
| LargeKitchenA | 91.20 | 89.87 | 92.00 | 90.13 | 91.73 | **92.80** | 90.40 | 90.13 | 91.82 |
| Lightning2 | 72.13 | 77.05 | 77.05 | 75.41 | 78.69 | 74.86 | **85.25** | **85.25** | **85.25** |
| Lightning7 | 80.82 | 82.19 | 69.86 | 80.82 | 82.19 | 76.26 | 79.45 | 83.56 | **84.02** |
| Mallat | 96.38 | 97.14 | 97.92 | 96.97 | 96.89 | 97.43 | 97.40 | 96.84 | **98.21** |
| Meat | 91.67 | 91.67 | 90.00 | 98.33 | 96.67 | 91.11 | 98.33 | **100.00** | 95.00 |
| MedicalImages | 77.89 | 78.16 | 78.33 | 78.95 | 79.61 | 78.95 | 80.53 | 81.32 | **81.80** |
| MiddlePOAG | 52.60 | 52.60 | **59.96** | 57.14 | 58.44 | 55.19 | 50.00 | 57.14 | 52.81 |
| MiddlePOC | 82.47 | **84.88** | 81.90 | 83.51 | 83.51 | 83.96 | 84.19 | 81.79 | 81.90 |
| MiddlePTW | 50.00 | 50.00 | 51.30 | 48.70 | 51.30 | 54.11 | 49.35 | **54.55** | 52.60 |
| MoteStrain | 94.17 | **94.49** | 93.58 | 93.45 | 93.85 | 93.08 | 90.18 | 91.45 | 90.23 |
| NonIFECGT1 | **96.54** | **96.54** | 95.73 | 94.86 | 95.37 | 96.01 | 95.62 | 95.78 | 95.84 |
| NonIFECGT2 | 95.73 | 95.67 | 95.81 | 94.91 | 95.11 | 95.91 | 95.93 | 96.03 | **96.56** |
| OliveOil | 80.00 | 83.33 | 85.56 | **86.67** | **86.67** | 82.22 | **86.67** | **86.67** | **86.67** |
| OSULeaf | 97.52 | 97.93 | 98.62 | 98.76 | 98.76 | **99.45** | 95.04 | 95.04 | 99.17 |
| PhalangesOC | 83.10 | 84.62 | 85.47 | 84.62 | 84.62 | 85.47 | 85.43 | **86.01** | 85.16 |
| Phoneme | 32.49 | 31.80 | 36.69 | 34.12 | 34.65 | **37.90** | 31.54 | 31.86 | 36.34 |
| Plane | **100.00** | **100.00** | **100.00** | **100.00** | **100.00** | **100.00** | **100.00** | **100.00** | **100.00** |
| ProximalPOAG | 84.88 | 85.85 | **87.15** | 86.83 | 85.85 | 85.53 | 86.34 | 86.83 | 86.34 |
| ProximalPOC | 91.41 | 92.44 | 91.29 | 89.69 | 91.41 | 90.61 | 92.78 | **93.81** | 93.70 |
| ProximalPTW | 76.59 | 77.07 | **81.79** | 77.56 | 77.56 | 79.84 | 76.59 | 78.05 | 75.93 |
| RefrigerationD | 52.00 | 49.87 | 54.84 | 52.27 | 51.73 | **58.49** | 51.20 | 52.53 | 54.13 |
| ScreenType | 62.13 | 62.67 | 63.91 | **64.27** | 63.47 | 62.76 | 58.13 | 57.87 | 58.49 |
| ShapeletSim | 90.00 | 89.44 | 96.11 | 97.78 | 91.11 | **99.81** | 92.22 | 96.67 | 99.26 |
| ShapesAll | 89.50 | 90.83 | 90.94 | 92.67 | 93.17 | 92.39 | 93.00 | 93.83 | **94.28** |
| SmallKitchenA | 79.73 | 81.07 | 80.89 | 79.73 | **82.93** | 80.98 | 76.80 | 81.33 | 80.18 |
| SonyAIBORS1 | 97.17 | 94.84 | 96.45 | 97.50 | **98.50** | 94.73 | 98.17 | 94.84 | 96.56 |
| SonyAIBORS2 | 98.01 | 98.32 | **99.16** | 97.17 | 96.96 | 98.04 | 94.02 | 82.90 | 89.65 |
| StarLightC | 96.48 | 97.34 | 98.04 | 97.64 | 97.94 | 98.19 | 97.98 | 98.03 | **98.21** |
| Strawberry | 97.30 | 97.84 | 97.30 | 97.30 | 97.03 | 96.94 | **98.65** | 98.11 | 96.94 |
| SwedishLeaf | 97.28 | 96.96 | 98.03 | 96.64 | 96.80 | **98.29** | 96.32 | 97.12 | 97.49 |
| Symbols | 94.47 | 96.68 | 97.79 | 87.44 | 85.73 | **98.63** | 97.69 | 96.58 | 96.95 |
| SyntheticC | 99.00 | 96.00 | 97.78 | 99.67 | 99.33 | 99.78 | **100.00** | **100.00** | **100.00** |
| ToeS1 | 96.49 | 95.61 | 96.35 | 95.61 | 94.74 | **97.66** | 96.05 | 96.93 | 97.08 |
| ToeS2 | 91.54 | 90.00 | 90.00 | 88.46 | 89.23 | 93.33 | 93.08 | 90.77 | **94.10** |
| Trace | **100.00** | **100.00** | **100.00** | **100.00** | **100.00** | **100.00** | **100.00** | **100.00** | **100.00** |
| TwoLeadECG | **100.00** | **100.00** | 99.94 | **100.00** | 99.91 | 99.88 | 99.91 | 99.91 | 99.91 |
| TwoPatterns | 87.75 | 99.78 | 89.18 | **100.00** | **100.00** | 94.27 | **100.00** | **100.00** | **100.00** |
| UWaveGLAll | 81.91 | 93.22 | 94.59 | 87.41 | 92.13 | 94.59 | **95.78** | 95.42 | 93.37 |
| UWaveGLX | 75.99 | 80.43 | 84.25 | 79.68 | 79.62 | 83.33 | 83.25 | 82.33 | **84.59** |
| UWaveGLY | 65.52 | 74.96 | 77.42 | 70.02 | 74.29 | 77.54 | 76.86 | 77.64 | **79.79** |
| UWaveGLZ | 73.73 | 76.44 | 80.58 | 76.63 | 76.35 | 80.67 | 77.19 | 76.10 | 78.80 |
| Wafer | 99.68 | 99.72 | 99.86 | 99.90 | 99.81 | **99.94** | 99.82 | 99.81 | 99.87 |
| Wine | 81.48 | 75.93 | 75.93 | 81.48 | 74.07 | 67.90 | **88.89** | 81.48 | 86.42 |
| WordS | 57.68 | 57.68 | 60.19 | 64.58 | 63.01 | 65.99 | 76.65 | 75.08 | **76.75** |
| Worms | 76.62 | 75.32 | 78.35 | 75.32 | 79.22 | 82.25 | 81.82 | 77.92 | **84.42** |
| WormsTwoC | 74.03 | 75.32 | 80.52 | 77.92 | 75.32 | **83.98** | 75.32 | 75.32 | 81.39 |
| Yoga | 85.03 | 88.87 | 90.79 | 87.60 | 89.70 | 90.84 | 91.90 | 91.87 | **94.82** |
| ACSF1 | 89.67 | 89.67 | 90.33 | 90.67 | 90.67 | 90.00 | **94.33** | **94.33** | 90.67 |
| BME | 66.44 | 99.78 | 99.78 | 85.33 | **100.00** | **100.00** | **100.00** | **100.00** | **100.00** |
| Chinatown | 96.99 | 97.76 | **98.25** | 97.57 | 97.86 | 97.96 | 97.86 | 97.86 | 98.06 |
| Crop | 73.63 | 75.93 | 76.61 | 75.31 | 77.38 | 78.03 | 78.34 | 78.35 | **78.54** |
| EOGHS | 64.27 | 64.46 | 64.73 | 66.48 | **66.57** | 65.38 | 65.01 | 64.92 | 65.19 |
| EOGVS | 35.54 | 43.46 | 40.88 | 42.17 | 43.65 | 40.52 | 47.15 | **47.79** | 39.78 |
| EthanolL | 44.40 | 78.93 | 81.07 | 68.33 | 84.33 | 79.67 | 84.27 | **85.60** | 78.73 |
| FreezerRT | 99.22 | 99.77 | 99.59 | **99.82** | 99.77 | 99.63 | 99.64 | 99.63 | 99.60 |
| FreezerST | 62.07 | 79.79 | 79.74 | 60.80 | 80.89 | 80.13 | 94.88 | **96.20** | 95.98 |
| GunPointAS | 98.63 | 99.68 | **100.00** | 99.68 | **100.00** | **100.00** | 99.68 | 99.58 | 99.37 |
| GunPointMVF | 99.58 | **99.68** | **99.68** | 99.37 | **99.68** | **99.68** | 99.47 | 99.58 | **99.68** |
| GunPointOVY | 87.09 | 99.68 | **100.00** | 90.26 | 99.68 | **100.00** | 98.52 | 98.31 | 99.68 |
| HouseT | **98.32** | **98.32** | **98.32** | 93.00 | 98.04 | **98.32** | **98.32** | **98.32** | 97.48 |
| InsectEPGRT | 99.73 | **100.00** | **100.00** | **100.00** | **100.00** | **100.00** | **100.00** | **100.00** | **100.00** |

Table A.8: Classification accuracy on 112 UCR datasets.

| DataSets | GAP -FCN | MIL -FCN | SuperMIL -FCN | GAP -ResNet | MIL -ResNet | SuperMIL -ResNet | GAP -IT | MIL -IT | SuperMIL -IT |
|---|---|---|---|---|---|---|---|---|---|
| InsectEPGST | 84.34 | 99.46 | 98.80 | 93.57 | **99.73** | 96.79 | 90.09 | 93.98 | 95.31 |
| MixedShapesRT | 94.96 | 96.92 | 97.66 | 97.57 | 98.08 | 98.02 | 97.79 | 97.84 | **98.74** |
| MixedSST | 90.86 | 90.86 | 94.98 | 93.94 | 93.94 | **95.82** | 91.48 | 91.48 | 95.77 |
| PigAP | 21.31 | 18.75 | 15.54 | 18.75 | 18.75 | 11.38 | **68.43** | 64.26 | 65.87 |
| PigAP | 98.24 | 98.24 | 99.20 | 98.40 | 98.40 | 99.68 | 99.20 | 99.20 | **100.00** |
| PigCVP | 88.62 | 88.62 | 88.94 | 91.51 | 91.51 | 92.15 | 95.83 | 95.83 | **97.28** |
| PowerC | 82.78 | 95.56 | 92.22 | 80.93 | 94.63 | 91.48 | 94.81 | **97.41** | 94.44 |
| Rock | 62.00 | 75.33 | 77.33 | 72.00 | 80.67 | 80.67 | 79.33 | 80.00 | **83.33** |
| SemgHandGC | 80.28 | 86.44 | 82.50 | 79.11 | 85.44 | 75.72 | 85.83 | **88.28** | 78.28 |
| SemgHandMCh2 | 43.04 | 42.81 | 30.44 | 40.30 | 46.44 | 38.67 | 45.48 | **47.26** | 32.74 |
| SemgHandSCh2 | 73.04 | 77.70 | 61.78 | 70.07 | 77.93 | 57.78 | 78.44 | **78.74** | 66.52 |
| SmoothS | 96.44 | 96.44 | 98.44 | 98.67 | **99.11** | 98.67 | **99.11** | **99.11** | 98.89 |
| UMD | **99.31** | **99.31** | **99.31** | **99.31** | **99.31** | **99.31** | **99.31** | **99.31** | **99.31** |
| Mean accuracy | 81.89 | 84.09 | 84.24 | 83.62 | 84.88 | 85.13 | 86.01 | 86.32 | **86.79** |
| Mean rank | 7.00 | 5.77 | 5.17 | 5.62 | 4.96 | 4.29 | 4.61 | 4.11 | **3.47** |

Table A.9: Classification accuracy on 26 UEA datasets.

| DataSets | GAP-FCN | MIL-FCN | SuperMIL-FCN | GAP-ResNet | MIL-ResNet | SuperMIL-ResNet | GAP-IT | MIL-IT | SuperMIL-IT | GAP-ConvT | MIL-ConvT | SuperMIL-ConvT | GAP-TodyN | MIL-TodyN | SuperMIL-TodyN |
|---|---|---|---|---|---|---|---|---|---|---|---|---|---|---|---|
| AWR | 98.67 | 98.33 | 98.89 | 98.33 | 98.56 | 99.00 | 98.89 | 98.56 | **99.11** | 98.22 | 97.78 | 98.22 | 98.44 | 98.67 | 98.33 |
| AtrialF | 6.67 | 11.11 | 6.67 | 15.56 | 6.67 | 13.33 | 13.33 | 6.67 | 8.89 | 15.56 | 20.00 | 20.00 | 35.56 | **37.78** | 26.67 |
| BM | **100.00** | **100.00** | **100.00** | **100.00** | **100.00** | **100.00** | **100.00** | **100.00** | **100.00** | **100.00** | **100.00** | **100.00** | **100.00** | **100.00** | **100.00** |
| Cricket | 98.61 | 98.61 | **99.07** | 98.61 | 98.61 | 98.61 | 98.61 | 98.61 | **99.07** | 98.61 | 98.61 | 98.61 | 98.61 | 98.61 | **99.07** |
| DuckDG | 64.00 | 62.00 | 67.33 | 61.33 | 62.67 | 65.33 | 65.33 | 65.33 | 69.33 | 70.67 | 69.33 | **73.33** | 64.67 | 60.00 | 66.67 |
| ERing | 88.52 | 88.77 | 88.89 | 90.49 | 89.51 | 90.37 | 93.70 | 93.83 | 94.07 | 92.22 | 92.84 | 92.10 | 89.63 | 93.21 | 89.26 |
| EW | 93.13 | 92.88 | 94.91 | 94.91 | **95.93** | 95.17 | 92.11 | 92.37 | 95.42 | 89.57 | 88.80 | 90.84 | 90.59 | 91.35 | 92.88 |
| Epilepsy | 99.28 | 99.28 | **100.00** | 97.83 | 98.31 | **100.00** | 97.10 | 97.10 | **100.00** | 97.83 | 98.31 | 98.79 | 96.14 | 94.44 | 96.38 |
| EthanolIC | **30.54** | 25.60 | 25.86 | 25.22 | 24.21 | 26.11 | 23.83 | 24.21 | 25.98 | 27.25 | 27.00 | 29.40 | 28.52 | 26.62 | 30.42 |
| FD | 55.31 | 60.81 | 57.07 | 55.46 | 63.65 | 58.60 | 67.49 | 67.57 | **68.07** | 66.61 | 66.12 | 66.13 | 66.19 | 66.28 | 63.54 |
| FM | 52.67 | 53.33 | 53.67 | 54.33 | 51.67 | 50.00 | 55.33 | 51.67 | 53.33 | 52.00 | 50.67 | 53.00 | 52.67 | 49.00 | **56.00** |
| HMD | 39.19 | 36.04 | 38.29 | 35.59 | 32.43 | 34.68 | 32.88 | 32.88 | 37.84 | 39.19 | 43.24 | 40.09 | 49.10 | 49.55 | **50.00** |
| HW | 57.84 | 57.49 | 50.00 | 62.00 | 63.49 | 54.12 | **71.61** | 71.45 | 71.02 | 50.86 | 53.37 | 52.27 | 55.53 | 55.73 | 58.78 |
| HB | 71.54 | 70.89 | 74.63 | 70.41 | 72.68 | **75.77** | 71.87 | 69.92 | 74.47 | 70.08 | 70.73 | 69.92 | 71.87 | 72.68 | 74.96 |
| LSST | 72.59 | 72.52 | 73.22 | 73.11 | 72.61 | **73.71** | 72.52 | 72.36 | 71.60 | 72.82 | 72.52 | 72.11 | 67.68 | 68.53 | 68.75 |
| Libras | 96.30 | 96.30 | 96.67 | 96.67 | 96.48 | **97.04** | 92.04 | 91.85 | 91.85 | 92.22 | 90.93 | 92.41 | 84.07 | 85.37 | 87.04 |
| MI | 52.33 | 54.00 | 53.00 | 55.33 | 55.67 | 50.67 | 52.67 | 51.67 | 54.33 | **59.00** | 58.67 | 57.33 | 53.33 | 48.33 | 51.67 |
| NATOPS | 97.04 | 97.22 | 95.19 | 96.85 | 97.04 | 97.41 | 97.59 | **99.07** | 97.78 | 96.85 | 96.85 | 97.59 | 93.89 | 95.37 | 93.70 |
| PEMS-SF | 79.19 | 79.19 | 76.88 | 78.81 | 79.38 | 77.07 | 83.62 | **84.39** | 82.85 | 74.76 | 75.53 | 76.88 | 81.70 | 80.15 | 83.82 |
| PD | 99.16 | 99.17 | 99.33 | 99.17 | 99.20 | **99.38** | 98.99 | 99.21 | 99.30 | 99.24 | 99.23 | 99.08 | 98.59 | 98.57 | 98.52 |
| PS | 31.47 | 33.11 | 34.73 | 32.96 | 34.70 | **37.27** | 35.09 | 34.02 | 36.79 | 30.77 | 31.28 | 29.18 | 30.25 | 30.38 | 29.33 |
| RS | 83.11 | 85.09 | 84.43 | 83.33 | 83.99 | **89.04** | 83.55 | 82.02 | 84.65 | 85.09 | 85.09 | 85.96 | 83.77 | 83.33 | 82.68 |
| SelfRSCP1 | 81.46 | 86.23 | 83.96 | 82.03 | 86.69 | 75.77 | 85.67 | 87.26 | 82.25 | 88.28 | 89.31 | 88.74 | 88.85 | 89.31 | **89.99** |
| SelfRSCP2 | 50.74 | 48.33 | 51.67 | 51.11 | 49.07 | 53.33 | 50.37 | 53.15 | 47.22 | 54.26 | **55.00** | 52.78 | 51.85 | 52.59 | 54.63 |
| StandWJ | 46.67 | 46.67 | **53.33** | 46.67 | 46.67 | 48.89 | 51.11 | 48.89 | 48.89 | 46.67 | 44.44 | 48.89 | 51.11 | 35.56 | 46.67 |
| UWaveGL | 88.75 | 88.65 | 88.02 | 88.75 | 89.38 | 90.62 | 91.98 | 92.08 | **93.02** | 90.83 | 91.46 | 90.31 | 86.04 | 88.65 | 89.69 |
| Mean acc. | 70.57 | 70.83 | 70.99 | 70.96 | 71.12 | 71.20 | 72.20 | 71.77 | **72.58** | 71.52 | 71.81 | 72.08 | 71.87 | 71.16 | 72.29 |
| Mean rank | 9.10 | 8.98 | 7.37 | 8.87 | 8.33 | 6.67 | 7.31 | 7.98 | **5.42** | 7.81 | 8.02 | 7.67 | 9.08 | 9.54 | 7.87 |