# OpenReview forum: "SuperMIL: Supervised Multiple Instance Learning for Time Series Classification"
_ICLR.cc/2026/Conference — Submitted to ICLR 2026_

### Official Review · Reviewer_W8AH · 2025-10-28

**Soundness:** 3
**Presentation:** 3
**Contribution:** 2
**Rating:** 4
**Confidence:** 4

**Summary:**

This paper introduces Supervised MIL (SuperMIL) for time series classification (TSC) that addresses two critical limitations of existing MILs for TSC: i) degraded instance-level feature representations and ii) inability in capturing temporal dependencies among instances. SuperMIL tackles these two challenges by proposing an iterative, dual-path optimization framework that jointly optimizes instance-level pseudo-supervision and bag-level weak supervision. It also introduces a `Hawkes pooling` module to capture local temporal dependencies and a `coupled multi-instance loss` to align the learning objectives of both instances and bags. The authors conducted experiments on the UCR and UEA benchmarks, comparing SuperMIL against standard Global Average Pooling (GAP) and traditional MIL frameworks using identical backbone models (e.g., FCN, ResNet, InceptionTime).

**Strengths:**

- The proposed HAWKES pooling is interesting. It leverages the well-known Hawkes process to capture the causality among time-series instances.
- The empirical results are encouraging, outperforming conventional MIL and GAP.
- The paper is easy to follow.
- The code is provided.

**Weaknesses:**

- The exploration of MIL for time-series classification (TSC) has been explored (see e.g., MIL-Net [1] and TimeMIL [2]). In both works, the authors have explored the limitation of "standard" MIL for TSC. For example, MIL-Net proposes a conjunctive pooling and TimeMIL models temporal dependencies as self-attention with a specially designed wavelet positional encoding. These closely related works are not compared. It seems to me that the performance of the proposed SuperMIL is lower than that of TimeMIL in UEA benchmarks (avg. accuracy: 72.58 vs 77.4). This largely reduces the readers' enthusiasm and weaken the main argument of the paper.

- Using pseudo instance labels from the instance-level importance scores has been widely explored in many previous MIL methods. This family of works resemble the knowledge distillation framework with dual streams (as used in this paper). In particular, DTFD-MIL has explored similar ideas of using the importance scores from one stream to improve the other stream in a distillation fashsion.

- The major difference between MIL from other supervised learning method is their inherent permutation-invariant assumption. The authors seem not discussing this point (as opposed to MIL-Net and TimeMIL), e.g. would the standard MIL assumption be violated by the proposed pooling method. Why MIL is a good candidate for TSC is not well-articulated.

- A direct comparison to other state-of-the-art TSC methods (e.g., PatchTST [3], ModernTCN [4]) should be conducted.

`References`

[1] Inherently Interpretable Time Series Classification via Multiple Instance Learning. ICLR'24

[2] TimeMIL: Advancing Multivariate Time Series Classification via a Time-aware Multiple Instance Learning. ICML'24

[3] A Time Series is Worth 64 Words: Long-term Forecasting with Transformers. ICLR'23

[4] ModernTCN: A Modern Pure Convolution Structure for General Time Series Analysis. ICLR'24

[5] DTFD-MIL: Double-Tier Feature Distillation Multiple Instance Learning for Histopathology Whole Slide Image Classification. CVPR'23

**Questions:**

- How the attention visualization (Fig. A.8) is calculated in the proposed Hawkes pooling ?
- I would suggest the authors to have a separate `related work` section to discuss closely related works in detail.

---

### Official Review · Reviewer_SZYe · 2025-10-30

**Soundness:** 2
**Presentation:** 3
**Contribution:** 2
**Rating:** 4
**Confidence:** 4

**Summary:**

This paper proposes SuperMIL, a supervised multi-instance learning (MIL) framework for multivariate time-series classification. Unlike conventional MIL, which only optimizes bag-level objectives, SuperMIL introduces instance-level pseudo supervision derived from bag labels and couples it with a Hawkes-inspired pooling mechanism to capture local causal dependencies between instances. A Coupled Multi-Instance Loss (CML) is further proposed to align instance- and bag-level learning. Experiments on several UEA/UCR datasets show consistent performance gains over standard MIL and attention pooling methods.

**Strengths:**

The paper addresses a real limitation in time-series MIL — the degradation of instance-level representations due to weak bag supervision — and attempts to alleviate it with additional supervision and structured pooling.

The combination of pseudo-supervised instance learning, Hawkes-inspired pooling, and coupled loss design forms a logically unified pipeline. The idea of introducing causality-inspired pooling into MIL is conceptually interesting.

The proposed SuperMIL improves accuracy and interpretability on several benchmarks compared to DSMIL and ABMIL. The Hawkes pooling mechanism provides a plausible way to model local dependencies.

Figures and algorithm descriptions are clear. The paper’s motivation and module relations are easy to follow.

**Weaknesses:**

The core components—pseudo supervision and attention-based pooling—are not new. The so-called “Hawkes Pooling” borrows from the idea of directional attention and event excitation but is not a true probabilistic Hawkes process. The proposed framework is an incremental integration rather than a fundamentally new MIL paradigm.

The instance-level pseudo labels are directly inherited from bag labels without external signals or confidence calibration. As a result, the “supervision” may introduce noise and is unlikely to significantly enhance instance discriminability.

The paper does not thoroughly compare pseudo supervision with standard MIL, nor analyze the contribution of the Hawkes pooling versus conventional attention pooling. The marginal performance gains (~1%) make it difficult to attribute the improvement to the proposed causal modeling.

The use of the term “causal” is loose — the Hawkes pooling mechanism models directional similarity, not structural causality. There is no causal validation or interpretability evidence supporting the claimed causal dependencies.

The code repository is empty, making it hard to verify implementation details and experimental claims.

**Questions:**

See weakness.

---

### Official Review · Reviewer_Hkos · 2025-11-01

**Soundness:** 3
**Presentation:** 3
**Contribution:** 2
**Rating:** 2
**Confidence:** 5

**Summary:**

This paper proposes SuperMIL (Supervised Multiple Instance Learning), a novel framework for time series classification that addresses two major limitations of traditional multiple instance learning (MIL) methods: weak instance-level feature learning and the lack of temporal causality modeling. SuperMIL introduces an iterative co-optimization mechanism that alternately refines pseudo-supervised instance learning (derived from bag-level labels) and weak bag-level supervision, enhancing both local discriminability and global robustness. It further incorporates two key components: a Hawkes Pooling module, which models causal dependencies between temporal instances by decomposing excitations into directional similarities and instance differences, and a Coupled Multi-Instance Loss, which aligns the objectives of instance- and bag-level representations through a global-local consistency constraint. Extensive experiments on the UCR and UEA time series datasets demonstrate that SuperMIL consistently outperforms traditional MIL and GAP baselines across multiple backbone networks in terms of classification accuracy, interpretability, and convergence efficiency, showing strong generalization ability and practical potential.

**Strengths:**

(1). This method provides gradient-level analysis demonstrating how delayed pooling (as in SuperMIL) preserves richer feature information than early pooling (in GAP/MIL)

(2). Extensive evaluation across >130 datasets (UCR/UEA), multiple architectures, and ensemble settings. Includes ablation, perturbation, and efficiency studies.

(3). The framework provides instance-level interpretability via attention and causal modeling

**Weaknesses:**

(1). Questionable novelty and missing references
The originality of SuperMIL is not fully convincing. The paper does not cite or compare with several recent and highly relevant works[1,2]. it is difficult to assess whether SuperMIL provides a substantial advancement over existing supervised or attention-based MIL methods.

[1]. Chen, Xiwen, et al. "TimeMIL: Advancing multivariate time series classification via a time-aware multiple instance learning." arXiv preprint arXiv:2405.03140 (2024).

[2]. Jang, Jaeseok, and Hyuk-Yoon Kwon. "TAIL-MIL: Time-aware and instance-learnable multiple instance learning for multivariate time series anomaly detection." Proceedings of the AAAI Conference on Artificial Intelligence. Vol. 39. No. 17. 2025.

[3]. Multiple Instance Learning for Efficient Sequential Data Classification on Resource-constrained Devices



(2) Lack of attention-based and anomaly detection validation
MIL is typically a weakly supervised framework that infers instance-level relevance from bag-level labels, where attention mechanisms play a central role. The paper does not discuss or compare with attention-based MIL techniques. Furthermore, validation on anomaly or abnormal detection tasks would strengthen the paper by demonstrating the framework’s ability to identify key instances and improving its practical applicability.

(3) Unclear treatment of multi-class classification
MIL is mainly formulated for binary classification, where a bag is labeled positive if at least one instance is positive. However, the datasets used in this paper are multi-class. The authors do not specify how this issue is handled. If a one-vs-rest binary formulation is used, it should be stated explicitly. If standard cross-entropy loss is directly applied, the model effectively becomes a conventional supervised classifier rather than a genuine MIL framework.

(4) Permutation invariance versus temporal dependency
A key property of MIL is permutation invariance, meaning that the model’s output should not depend on the order of instances. However, temporal dependency is critical in time series data. The paper does not clearly explain how SuperMIL balances this conflict or whether the proposed Hawkes Pooling mechanism maintains temporal order while preserving the theoretical properties of MIL[3,,4].

[3]. Ilse, Maximilian, Jakub Tomczak, and Max Welling. "Attention-based deep multiple instance learning." International conference on machine learning. PMLR, 2018.

[4]. Qi, Charles R, Su, Hao, Mo, Kaichun, and Guibas, Leonidas J. PointNet: Deep learning on point sets for 3d classification and segmentation. In CVPR, 2017.

**Questions:**

I have serious doubts about whether the proposed SuperMIL framework can be considered a genuine MIL method, especially given its application to multi-class classification tasks. The paper does not clearly explain how the MIL formulation is preserved in this setting. It would be important to reference and follow the standard validation procedures used in previous MIL works to clarify whether SuperMIL truly adheres to the MIL paradigm.[1,2]

[1] Are Multiple Instance Learning Algorithms Learnable for Instances? NeurIPS, 2022.

[2] Raff E., & Holt J. Reproducibility in Multiple Instance Learning: A Case for Algorithmic Unit Tests. NeurIPS, 2023.

---

### Official Review · Reviewer_ggah · 2025-11-01

**Soundness:** 2
**Presentation:** 2
**Contribution:** 2
**Rating:** 2
**Confidence:** 3

**Summary:**

This paper proposes SuperMIL, a supervised multiple instance learning (MIL) framework for time series classification (TSC), introducing a dual-optimizer iterative training strategy, a Hawkes pooling module for causal modeling, and a coupled multi-instance loss for aligning bag- and instance-level objectives. The method  shows improvements on UCR/UEA benchmarks.

**Strengths:**

1. The paper introduces a  supervised MIL framework that combines instance-level pseudo-supervision with bag-level weak supervision in an iterative fashion.
2. The use of Hawkes pooling effectively captures local temporal causality and improves the quality of instance-level representations.

**Weaknesses:**

1. The proposed dual-optimizer iterative training increases computational cost and complicates convergence analysis. The paper lacks sufficient analysis of training time and stability, especially for long sequences.
2. The instance-level pseudo-supervision relies heavily on inherited bag labels. While iteration helps reduce noise, there is no explicit mechanism for pseudo-label refinement or confidence calibration.
3. The paper misses discussion and comparison with several recent MIL and TSC approaches such as TimeMIL [1], MILLET [2], and strong TSC baselines like [3] [4]. The baselines used in the paper are outdated.

[1] TimeMIL: Advancing Multivariate Time Series Classification via a Time-aware Multiple Instance Learning. ICML'25

[2] INHERENTLY INTERPRETABLE TIME SERIES CLASSIFICATION VIA MULTIPLE INSTANCE LEARNING. ICLR'24

[3]  Learning soft sparse shapes for efficient time-series classification. ICML'25

[4] ModernTCN: A modern pure convolution structure for general time series analysis. ICLR'24

**Questions:**

See the weaknesses for details.

---

### Meta-Review · Area_Chair_6XDH · 2025-12-17

**Summary:**

This paper presents a well-motivated and clearly explained framework that integrates pseudo-supervised instance learning with a novel Hawkes pooling mechanism for time-series MIL. While reviewers found the approach logical and noted its improved interpretability over baselines, major concerns must be addressed. First, the novelty is considered incremental, as core components resemble existing ideas, and crucial comparisons with recent state-of-the-art methods (e.g., TimeMIL, modern TSC models) are missing. Second, key methodological issues include the unrefined reliance on bag-derived pseudo-labels, the unclear handling of multi-class MIL, and the unresolved tension between permutation invariance and temporal order. Furthermore, the modest performance gains, loose use of "causality," and lack of code availability weaken the contribution. A substantial revision addressing these points through expanded comparisons and deeper analysis is necessary to demonstrate a significant advance.

**Reviewer Concerns:**

There is no rebuttal from the authors. Therefore, all concerns are still outstanding.

**Reviewer Scores:**

There is no discussion between reviewers and authors due to no rebuttal.

---

### Decision · Program_Chairs · 2026-01-26

Reject